

**Personal exposure to PM$_{2.5}$ emitted from typical anthropogenic sources in**

**Southern West Africa (SWA): Chemical characteristics and associated**

**health risks**

Hongmei Xu[1*,2,3,4], Jean-François Léon[2], Cathy Liousse[2*], Benjamin Guinot[2], Véronique

Yoboué[5], Aristide Barthélémy Akpo[6], Jacques Adon[2], Kin Fai Ho[7], Steven Sai Hang Ho[3],

Lijuan Li[3], Eric Gardrat[2], Zhenxing Shen[1], Junji Cao[3]

*[1]Department of Environmental Science and Engineering, Xi'an Jiaotong University, Xi'an,*
*China*
*[2]Laboratoire d'Aérologie, Université de Toulouse, CNRS, Toulouse, France*
*[3]SKLLQG, Key Lab of Aerosol Chemistry & Physics, Institute of Earth Environment, Chinese*
*Academy of Sciences, Xi'an, China*
*[4]Collaborative Innovation Center of Atmospheric Environment and Equipment Technology,*
*Jiangsu Key Laboratory of Atmospheric Environment Monitoring and Pollution Control*
*(AEMPC), Nanjing University of Information Science & Technology, Nanjing, China*
*[5]Laboratoire de Physique de l'Atmosphère, Université Felix Houphouet-Boigny, Abidjan,*
*Côte d'Ivoire*
*[6]Laboratoire de Physique du Rayonnement, Université Abomey-Calavi, Abomey-Calavi,*
*Bénin*
*[7]JC School of Public Health and Primary Care, The Chinese University of Hong Kong, Hong*
*Kong, China*
*Corresponding authors:*
*Hongmei Xu, E-mail:* xuhongmei@xjtu.edu.cn
*Cathy Liousse, E-mail:* cathy.leal-liousse@aero.obs-mip.fr



**Abstract**

Urbanization is a strongly emerging issue in Southern West African (SWA) region.

There is a general lack of understanding about the personal exposure to fine particulate matter

($PM_{2.5}$), its chemical components and health risks related to the various anthropogenic

sources in this region. In the current study, personal exposure to $PM_{2.5}$ (PE $PM_{2.5}$) sampling

was for the first time carried out in dry season (January) and wet season (July) of 2016 to

characterize PE $PM_{2.5}$ from Domestic Fires (DF) for women and Waste Burning (WB) for

students in Abidjan, Côte d'Ivoire and Motorcycle Traffic (MT) for drivers in Cotonou, Benin.

The average PE $PM_{2.5}$ mass concentrations were 331.7±190.7, 356.9±71.9 and

242.8±67.6 μg m$^{-3}$ at DF, WB and MT for women, students and drivers, which were 2.4, 10.3

and 6.4 times of the ambient $PM_{2.5}$ concentrations, respectively. Mean concentrations of PE

$PM_{2.5}$ at DF (358.8±100.5 μg m$^{-3}$), WB (494.3±15.8 μg m$^{-3}$) and MT (335.1±72.1 μg m$^{-3}$)

were much elevated in dry season, 15% higher than that at DF and 55% higher at both WB

and MT. The changes in PE $PM_{2.5}$ can be attributed to the source emissions, meteorological

factors and personal activities. The results also show that geological material (35.8%, 46.0%

and 42.4%) and organic matter (34.1%, 23.3% and 24.9%) were always the major

components in PE $PM_{2.5}$ at DF, WB and MT sites. It is worth noting that the contribution to

PE $PM_{2.5}$ from heavy metals was higher at WB (1.0%) than at DF (0.7%) and MT (0.4%),

which was influenced by the waste burning emission strongly, leading to the highest heavy

metal non-cancer risks for students (5.1 and 4.8 times of women and drivers' non-cancer

risks).

In organic species of PE $PM_{2.5}$, some fingerprints can be used to quantify the exposure

concentrations and trace the source contributions from local typical anthropogenic sources to

different samples. Women exposure concentration to polycyclic aromatic hydrocarbons

(PAHs) in $PM_{2.5}$ at DF (77.4±47.9 ng m$^{-3}$) was 1.6 times that for students at WB (49.9±30.7

53    ng m$^{-3}$) and 2.1 times for drivers at MT (37.0±7.4 ng m$^{-3}$), which is related to the solid fuels

burning and grilling meat activities, resulting in 5 times higher of cancer risk safety threshold

($1×10^{-6}$) to women. Phthalate esters (PAEs), commonly used as plasticizers in many products,

were observed to be extremely high in student exposure $PM_{2.5}$ samples (1380.4±335.2 ng m$^{-3}$)

at WB site, owing to the waste burning emission obviously. Drivers exposure to fossil fuel

emission (especially traffic) markers-hopanes in PE $PM_{2.5}$ at MT (50.9±7.9 ng m$^{-3}$) was 2.0-

2.3 times higher than women at DF (17.1±6.4 ng m$^{-3}$) and students at WB (15.6±6.1 ng m$^{-3}$),

correlating with the elevated exposure to traffic emissions for drivers.

Overall, the study shows that wood combustion, waste burning, fugitive dust and motor



vehicle emissions dominated PE PM$_{2.5}$ mass and contributed to its toxicities mainly. Heavy

metals and organic chemicals in PE PM$_{2.5}$ in SWA brought about Pb and Mn non-cancer

health risks for students at WB site and serious PAHs cancer risks for women at DF site via

inhalation pathway. This study provides basic data and initial perspective of PM$_{2.5}$ personal

exposure and health risk assessment in underdeveloped area to encourage the government to

improve the air quality and living standard of residents in this region.

***Keywords:*** personal exposure to PM$_{2.5}$; domestic fires; waste burning; motorcycle traffic;

West Africa



## 1. Introduction

The southern West Africa (SWA) region has experienced an economic upturn and

increasingly significant anthropogenic air pollutant emissions during the last few years and

causes serious air pollution (IMF, 2017; Norman et al., 2007). Fine particulate matter ($PM_{2.5}$

with equivalent aerodynamic diameters $\leq 2.5$ μm) is one of the major concerns of

international organizations and publics because of the health effects associated with exposure

levels, health of individuals and pollutant emission sources (Bruce et al., 2000; Chen et al.,

2013; Owili et al., 2017). Owili et al. (2017) found that the four types of ambient $PM_{2.5}$,

including mineral dust, anthropogenic pollutant, biomass burning and mixture aerosols are

significantly associated with under-five and maternal mortality in Africa. However, studies

on $PM_{2.5}$, especially direct personal exposure to $PM_{2.5}$ (not stationary sampling) and its health

effects are still limited in low income countries in this region.

Since the 1990s, several international campaigns have been performed in Africa. Some

of them were mainly focused on the particles or aerosols, for example DECAFE (Lacaux et

al., 1995), EXPRESSO (Delmas et al. 1999; Ruellan et al., 1999), SAFARI-1992 (Lindesay et

al., 1996), SAFARI-2000 (Swap et al., 2002), AMMA (Léon et al., 2009; Liousse et al., 2010;

Marticorena et al., 2010) and INDAAF (Ouafo-Leumbe et al., 2017). As we known, the

Africa is the largest source of mineral dust particles from the Sahara Desert and unpaved road

surfaces (Laurent et al., 2008; Marticorena et al., 2010; Reeves et al., 2010), carbonaceous

aerosols originated from wild fires (mainly savannah fires) as well (Capes et al., 2008;

Gaudichet et al., 1995). Therefore, these campaigns were more biased towards the natural

sources of aerosols in Africa. Liousse et al. (2014) have shown the increase of the relative

importance of particulate emissions from domestic fires and fossil fuel combustion in Africa.

In previous literature, the major contributions to the aerosol chemistry in the dry season in

northern Benin were dust (26%‐59%), primary organic matters (POC, 30%‐59%), elemental

carbon (EC, 5%‐9%) and water soluble inorganic ions (3%‐5%) (Ouafo-Leumbe et al., 2017).

This poses serious health questions for people who frequent the city on a daily basis.

However, there is still limited literature on the health effects of personal exposure to $PM_{2.5}$

emitted from the typical anthropogenic sources in the emerging cities in Africa.

The main anthropogenic emission sources of $PM_{2.5}$ in SWA include domestic wood

burning, fossil fuel combustion, unregulated traffic and industries, waste burning and road

dust associated to human activities. An ongoing project in Africa-DACCIWA (Dynamics-

Aerosol-Chemistry-Cloud Interactions in West Africa) aims at quantifying the influence of

anthropogenic and natural emissions on the atmospheric composition over South West Africa



and assessing their impact on human, ecosystem health and agricultural productivity, which will be communicated to policy-makers, scientists, operational centres, students and general publics. The current work involved in the framework of the Work Package 2 "Air Pollution and Health" of DACCIWA is trying to link emission sources, air pollution and health impacts over representative differentiated urban sources: domestic fires and waste burning in Abidjan (Ivory Coast) and two-wheel vehicle traffic emission in Cotonou (Benin) for different target populations.

Smoking meat (fish and pork) by biomass fuels (mainly wood) is an important diet pattern for residents of coastal countries in SWA area. Many female workers without any personal health protection are engaged in roasting activity. They are directly exposed to extremely $PM_{2.5}$ pollution from wood burning and smoking meat, causing very serious health problems. Urbanization leads to explosive population growth and rural depopulation in SWA, resulting in a large amount of urban domestic waste. The biggest landfill in Abidjan focused in this study received more than 1,000,000 t waste per year (Adjiri et al., 2015). A mass of garbage lacks processing capacity and reasonable treatment method, resulting in a large amount of air pollutants emitted during the combustion and stacking of waste, which damages the living environment and health condition of the populations in Abidjan, especially for children (UNEP, 2015). Moreover, in many low-income countries, motorbike taxis are a major mode of local transportation (Assamoi and Liousse, 2010). In Benin, motorbike taxi drivers (mainly male) represented almost 2.5% of the total population of Benin in 2002 (Lawin et al., 2016). As they spend many hours in the middle of traffic every day, these drivers are highly exposed to traffic-related $PM_{2.5}$ pollution over years.

Major chemical components in $PM_{2.5}$, like OC, ions and EC mentioned above, not only strong impact on $PM_{2.5}$ physicochemical characteristics, but also affect its health risks. Typical trace toxic chemicals, such as heavy metals and polycyclic aromatic hydrocarbons (PAHs) can be attached on $PM_{2.5}$, which would cause various health problems for humans (Cao et al., 2012; WHO, 1998; Xu et al., 2015). For example, Pb is neuro-developmental metal, affecting children health and mental development seriously (USEPA, 2006; Xu et al., 2017). PAHs can be teratogenic and carcinogenic for humans strongly (Tang et al., 2008). Up to now, only few studies have investigated $PM_{2.5}$ chemical compositions of the personal exposure samples, and little is known regarding the sources and health risks of personal exposure $PM_{2.5}$ in SWA region. This poses a challenge to formulation of strategies aimed at mitigating $PM_{2.5}$ pollution and its health effects in this area.

Therefore, our study relies on the portative device sampling $PM_{2.5}$ personal exposure





samples in SWA area in 2016 for the purpose of 1) characterizing the personal exposure to

$PM_{2.5}$ as variation of different local typical anthropogenic $PM_{2.5}$ sources by the chemical

component analysis and $PM_{2.5}$ mass balance; 2) identifying potential pollution sources to

different exposed populations by fingerprint organic markers; 3) evaluating the personal

exposure to $PM_{2.5}$ health risks by the U.S. EPA health risk assessment model. This

information will provide scientific understanding of the personal exposure to $PM_{2.5}$ in SWA

and try to arouse the government's attention to protect residents there from various

anthropogenic sources.

**2. Materials and methods**

*2.1. Site description and participants selection*

Personal exposure to $PM_{2.5}$ (hereafter defined as PE $PM_{2.5}$) filter samples were collected

using portative devices in the polluted atmosphere of different source environments,

including Domestic Fires (DF) for women, Waste Burning (WB) for students both in Abidjan,

Côte d'Ivoire, and Motorcycle Traffic (MT) for drivers in Cotonou, Benin (Figure 1). Abidjan

(5°20′ N, 4°1′ W) is the economic capital of Côte d'Ivoire with 6.5 million inhabitants in

2016. It is characterized by a high level of industrialisation and urbanization in SWA area.

Cotonou (6°21′ N, 2°26′ W) is the largest city and economic center of Benin, with about 1.5

million inhabitants in 2016. Both cities experience a tropical wet and dry climate, with

relatively constant air temperature (24-30 °C) and average relative humidity (RH) above 80%

throughout the year.

DF site in Abidjan is located in the market of Yopougon-Lubafrique (5°19.7′ N, 4°6.4′

162     W) in a large courtyard with about 25 fireplaces (Figure 2). The fuels used are essentially

hevea wood (one kind of rubber trees) locally. Several adult female workers were employed

to grilling meat or roasting peanuts from 06:00 to 15:00 UTC (working time) in the working

165     day. In this study, we selected two healthy, non-smoking female workers (average age of 32.5

166     years old) to investigate personal exposure to $PM_{2.5}$ from domestic fire and related sources,

such as grilling (Figure 2). WB site in Abidjan is near the public landfill of Akouédo (5°21.2′

168     N, 3°56.3′ W), which has received all the waste produced from Abidjan for the last 50 years

(Figure 2). We selected two healthy, non-smoking primary school students (average age of 11

170     years old) who live and study next to WB site (within 100 m straight-line distance) to

determine the personal exposure to $PM_{2.5}$ from waste burning (spontaneous combustion at

high air temperature condition and combustion by the landfill workers sometimes) emission

at landfill and other daily sources. MT site in Cotonou is located in the Dantokpa area (6°22.1′




174 N, 2°25.9′ E), one of the biggest markets in western Africa (Figure 2). It is largely dominated

by a mass of motorcycle traffic (two-wheel vehicles powered by petrol, also named zemidjan

in local language) and a small quantity of other motor vehicles emissions. We chose two

healthy, non-smoking male motorcycle drivers (average age of 50 years old) to survey $PM_{2.5}$

personal exposure from motorcycle emission and related sources (such as road dust).

Two women (woman A and B) involved in this study at DF are both in charge of

cooking at home by charcoal and butane gas, and cleaning house in daily life (Figure S1abc).

One of the student participators (student A, boy, 8 years old) at WB doesn't cook at home by

himself (energy sources for cooking are charcoal and liquefied petroleum gas (LPG)) (Figure

S1ac), but the other student (student B, girl, 14 years old) is usually responsible for cooking

at home by burning solid fuel, i.e., wood (Figure S1d). Two motorcycle drivers (driver A and

B) focused in this study at MT are both working for a local motorcycle operation company,

whose working time is usually from 6:30 to 10:30, 12:00 to 17:00 and 18:30 to 21:00 UTC.

They are driving on road almost all the working time and go back home for meals. They

don't cook at home by themselves (energy source for cooking is charcoal) (Figure S1a).

*2.2. Personal exposure to $PM_{2.5}$ samples collection and QA/QC*

12-hour integrated (daytime: 7:30 to 19:30 UTC; nighttime: 19:30 to 7:30 on the next

192 day UTC) PE $PM_{2.5}$ samples were collected during the dry season (from January $6^{th}$ to $11^{th}$)

and wet season (from July $5^{th}$ to $10^{th}$), 2016 in two major southwestern African cities

mentioned above (Figure 1). PE $PM_{2.5}$ sampling was conducted during three consecutive days

with the same type participants synchronously, using the PEM (Personal Environmental

Monitor) sampling devices with SKC pump (SKC Inc., USA) at a flow rate of 10 liter per

minute (lpm). The PEM $PM_{2.5}$ sampling head worn in the breathing zone of participants in

this study. Samples were collected on 37 mm pre-baked quartz filters (800 °C, 3 hours,

QM/A®, Whatman Inc., UK). A total of 72 personal exposure samples, including 24 samples

(12 daytime + 12 nighttime samples) for women at DF, 24 (12 + 12) for students at WB and

24 (12+12) for drivers at MT, were collected in this study. Moreover, 12 PE $PM_{2.5}$ field

blanks (one field blank for each participant in one season, collected on the second day of the

three consecutive sampling days) were sampled in this study as well.

In order to verify the comparability of personal exposure samples and data caused by not

identical sampling devices, 10 pairs of $PM_{2.5}$ samples were synchronously collected by two

sets of actual PEMs with SKC pumps. The comparison results led to a significant correlation

between the $PM_{2.5}$ mass concentrations obtained from two sampling devices





(y=0.986x+0.189, $R^2$=0.974, $P$<0.0001). Identical membrane (quartz fiber) and analytical

treatments were used in this study. After sampling, the filter samples were placed in Petri

dishes, sealed with parafilm and stored in a -20 °C freezer to prevent loss of mass through

volatilization prior to analysis. Blank values were used to account for any artifacts caused by

gas absorption and subtract the background $PM_{2.5}$ and chemical compositions concentrations

in this area.

We report the meteorological observations during the dry (December 2015 to March

2016) and wet (April to July 2016) seasons at the sampling places in Table 1. Meteorological

data are retrieved from the NOAA Global Surface Summary of the Day I (GSOD) at the

airports of each cities, namely Felix Houphouet Boigny Airport (Abidjan) and Cardinal

Bernadin Gantin International Airport (Benin). We give the daily average air temperature,

wind speed and rainfall accumulation in Table 1.

*2.3. $PM_{2.5}$ gravimetric and chemical analysis*

PE $PM_{2.5}$ filter samples were analyzed gravimetrically for mass concentrations with a

high-precision electronic microbalance (Sartorius MC21S, Germany) at Laboratoire

d'Aérologie (Toulouse, France) before and after sampling in the weighing room after

equilibration at 20-23 °C and the RH of 35%-45% for 24-hour. The absolute errors between

replicate weights were less than 0.015 mg for blank filters and 0.020 mg for sampled filters.

Total carbon (TC) was determined on 0.5 $cm^2$ punch-out of the filters by a carbon

analyzer (Ströhlein Coulomat 702C, Germany) at the Observatoire Midi-Pyrenees (OMP,

Toulouse, France). The quartz filter samples were subjected to a thermal pretreatment step

(kept at 60°C for 20 mins) in order to remove the volatile organic compounds (VOCs) and

eliminate water vapor. Subsequently the filters were combusted at 1200°C under $O_2$ and

detected as $CO_2$ in the carbon analyzer. Elemental carbon (EC) was obtained using a two-step

thermal method: step 1 consisted in a pre-combustion at 340 °C under $O_2$ for 2 h in order to

remove organic carbon (OC); step 2 consisted in the oxidation of the remaining EC at

1200 °C under $O_2$. The difference (TC-EC) yielded OC concentration (Benchrif et al., 2018;

Cachier et al., 2005).

To extract the water-soluble inorganic ions from the quartz filters, 1/4 of the filter was

placed in a separate 15 mL vials containing 10 mL distilled-deionized water (18.2 MΩ

resistivity). The vials were placed in an ultrasonic water bath and shaken with a mechanical

shaker for 45 min (15 min × 3 times) to extract the ions. The extracts were filtered through

0.45 μm pore size microporous membranes. After that, three anions ($Cl^-$, $NO_3^-$ and $SO_4^{2-}$) and



five cations (Na$^+$, NH$_4^+$, K$^+$, Mg$^{2+}$ and Ca$^{2+}$) in aqueous extracts of the filters were determined by an ion chromatograph (IC) analyzer (Dionex-600, Dionex, Sunnyvale, CA, USA), which was equipped with an AS11-HC anion column and a CS12 cation column for separation. Details of the IC method are described in Bahino et al. (2018) and Cachier et al. (2005).

One element: Fe (representing earth's crust emission) and ten heavy metals: V, Cr, Mn, Co, Ni, Cu, Zn, Sb, Ba and Pb in PE PM$_{2.5}$ samples were determined by Energy Dispersive X-Ray Fluorescence (ED-XRF) spectrometry (the PANalytical Epsilon 5 ED-XRF analyzer, the Netherlands) on 1/4 of filters in this study as well. The relative errors for all measured elements were < 6% between NIST Standard Reference Material (SRM) 2783 and our ED-XRF results, which is well within the required range of error, demonstrating the accuracy of ED-XRF. Replicate analysis of one quartz-fiber filter sample (five times) yielded an analytical precision between 5.2%-13.9%. Details of the ED-XRF measurements are described in Brouwer (2003) and Xu et al. (2012).

0.1-1.0 cm$^2$ punch-outs aliquots from the quartz filters were used to quantify the organic compounds, including polycyclic aromatic hydrocarbons (PAHs), phthalate esters (PAEs) and hopanes (see the specific organic species and their abbreviations measured in this study in Table 5) by an in-injection port thermal desorption-gas chromatography/mass spectrometry (TD-GC/MS) method. The approach has the advantages of shorter sample preparation time (< 1 min), minimizing of contaminations from solvent impurities, and higher sensitivity, compared with the traditional solvent extraction-GC/MS method. The detail analytical procedures have been reported in previous publications (Ho and Yu, 2004; Ho et al., 2008, 2011; Xu et al., 2013, 2016a). The results of the blank analyses showed only trace contamination levels (< 5.0%) of PE PM$_{2.5}$ samples concentrations.

*2.4. Health risk assessment model*

As we known, heavy metals and toxic organic species are associated with negative personal exposure health effects (Škrbic et al., 2016; Val et al., 2013; Wang et al., 2017a; Xu et al., 2018a). In this study, four heavy metals (Mn, Ni, Zn and Pb) and all PAHs and PAEs species in PE PM$_{2.5}$ were selected to determine the personal exposure inhalation health risks. The carcinogenic and non-carcinogenic health risks of PM$_{2.5}$ chemical species were calculated according to the U.S. EPA health risk assessment model (USEPA, 2004, 2011). The average daily exposure dose (D) via inhalation was estimated to assess the risk by the equations (1) as follows:





$$D = (C \times R \times EF \times ED \times cf) / (BW \times AT) \qquad (1)$$

the definitions and recommended values of parameters are shown in Table 2.

A hazard quotient (HQ) for non-cancer risk of heavy metals in PE $PM_{2.5}$ samples can be

obtained from equation (2):

$$HQ = D/RfD \qquad (2)$$

the threshold value of RfD indicates whether there is an adverse health effect during a certain

period. Hazard index (HI) can be obtained by summing up the individual HQ to estimate the

total non-cancer risks. If the HI < 1, then non-carcinogenic effect is impossible; HI ≥ 1,

adverse health effect might likely appear (Hu et al., 2012).

The incremental lifetime cancer risk (ILCR) of PAHs and PAEs in personal exposure

$PM_{2.5}$ samples can be calculated by multiplying the cancer slope factor (CSF) of PAHs and

PAEs with D as equation (3):

$$ILCR = D \times CSF \qquad (3)$$

for cancer risk, the value of $1 \times 10^{-6}$ is an internationally accepted as the precautionary or

threshold value above which the risk is unacceptable (Jedrychowski et al., 2015).

It is worth noting that, among the nineteen PAHs, BaP has been used as an indicator of

PAHs carcinogenicity (Wang et al., 2006). The carcinogenic health risk of PAH species can

be assessed by $[BaP]_{eq}$ instead (Yassaa et al., 2001) by equation (4):

$$\Sigma[BaP]_{eq} = \Sigma (C_i \times TEF_i) \qquad (4)$$

Besides, the carcinogenic risk for PAEs was assessed by DEHP, which is identified as a

possible carcinogen to humans by the International Agency for Research on Cancer (IARC)

(IARC, 1982; Li et al., 2016). The definitions and recommended values of the parameters in

equations (2-4) are also shown in Table 2 and Table 3.

*2.5. Questionnaire and time-activity diary*

Questionnaire (Appendix A-C) and time-activity diary (Appendix D) were collected

from each participant during the sampling period, respectively, to fully grasp the basic

information, potential exposure sources and activities of participants. In the questionnaire,

personal information, family status, dermatological, asthma symptoms, medical history,

current health status and so on were first asked from each participant. Besides, the questions

for women include: (1) living habits and environment (past and current living conditions,

general living habits, cooking habits and domestic fuel type/usage); (2) work environment

and travel habits (workplace, work nature, working time and daily travel mode/time); and (3)

affected by the burning of domestic solid fuels and roasting meat. The questions for students





include: (1) living habits and environment (past and current living conditions, general living

habits, participation in household duties, the family cooking habits and domestic fuel

type/usage; distance from home to WB site); (2) school environment and travel habits (school

location and related environment and daily travel mode/time); and (3) affected by the burning

of waste and household air pollution source. The questions for drivers include: (1) living

habits and environment (past and current living environments, general living habits,

participation in household duties, the family cooking habits and domestic fuel type/usage); (2)

work environment and travel habits (motorcycle power type, driving conditions, working

time and daily travel mode/time); and (3) affected by the motorcycle emission and household

air pollution source.

The time-activity diaries requested the participants to mark on half an hour basis

(sleeping time excluded) to assess each microenvironment time spending and detailed

activities.

### 3. Results and discussion

*3.1. Personal exposure to $PM_{2.5}$ and its chemical compositions*

*3.1.1. PE $PM_{2.5}$ mass concentration*

The average personal exposure to $PM_{2.5}$ (PE $PM_{2.5}$) mass concentrations were

$331.7 \pm 190.7$, $356.9 \pm 71.9$ and $242.8 \pm 67.6$ µg m$^{-3}$ for women at Domestic Fires (DF), students

at Waste Burning (WB) and drivers at Motorcycle Traffic (MT) respectively in 2016 in

Southern West Africa (SWA). Among these three types of subjects, the average

concentrations of PE $PM_{2.5}$ for women and students were quite similar, ~40% higher than the

drivers. PE $PM_{2.5}$ ranged from 106.2 µg m$^{-3}$ (nighttime in dry season, January 7$^{th}$) to 1164.7

333    µg m$^{-3}$ (daytime in wet season, July 5$^{th}$) for women at DF; from 37.8 µg m$^{-3}$ (nighttime in wet

season, July 8$^{th}$) to 1137.0 µg m$^{-3}$ (daytime in dry season, January 11$^{th}$) for students at WB;

and from 65.0 µg m$^{-3}$ (nighttime in wet season, July 11$^{th}$) to 648.5 µg m$^{-3}$ (daytime in dry

season, January 15$^{th}$) for drivers at MT. The ranges and standard deviations of PE $PM_{2.5}$

concentrations were extremely large, especially for women, because there are direct

combustion sources close around the women workers in this study. Moreover, the variations

of personal physics activities and air pollution source intensities lead to a drastic fluctuation

for PE $PM_{2.5}$.

The average mass concentrations of PE $PM_{2.5}$ were $358.8 \pm 100.5$, $494.3 \pm 15.8$ and

$335.1 \pm 72.1$ µg m$^{-3}$ in dry season (January), and $304.6 \pm 284.5$, $219.5 \pm 71.3$ and $150.6 \pm 10.4$ µg

m$^{-3}$ in wet season (July) for women at DF, students at WB and drivers at MT, respectively



(Table 4). Compared to dry season, the reduction rate of PE PM$_{2.5}$ for women at DF in wet

season was approximately 15%, while the sharp reductions were observed for students and

drivers at a similar level by more than 50%. PE PM$_{2.5}$ concentrations reducing could be

attributed to the occurrence of increased levels of rainfall in wet season in SWA (Table 1),

which causes the large reduction of road dust exposed to drivers and limits the garbage

spontaneous combustion significantly around students. Moreover, large scale transport of

mineral dust and combustion aerosols emitted by savannah wild fires contribute significantly

to the aerosol load during the dry season (Djossou et al., 2018), which is more important at

WB and MT than at DF (crowded community environment).

PE PM$_{2.5}$ mass concentrations in the daytime were much higher than those at night, no

matter in dry or wet season (Table 4 and Figure 3). The 12-hour averaged PE PM$_{2.5}$

concentrations show day/night (D/N) ratios of 3.4 (3.8 in dry season and 3.1 in wet season),

2.7 (2.8 and 2.5) and 2.4 (1.5 and 3.3) for women at DF, students at WB and drivers at MT,

respectively. Intensive human activities during the day, such as solid fuel combustion, waste

combustion or motor vehicle emission around the different group subjects, enhance the levels

of PM$_{2.5}$ exposure in the daytime. For example, lower PM$_{2.5}$ personal exposure level for

students at night at WB can be explained also by the fact that the participants in this study

usually spent most of the time indoors at night with limited physical activity, leading them to

be able to stay a distance and/or shelter from obvious emission sources (waste combustion)

outdoors. Besides, big fluctuations of D/N ratios for drivers were observed, with lower value

in dry season and higher in wet season. Relatively lower D/N ratio probably attributes to

nighttime driving (18:30 to 21:00 UTC) after dinner, which enhances their PM$_{2.5}$ exposed

levels from vehicle emission and road dust. Much higher D/N ratios in wet season attribute to

the increase in precipitation in wet season in Cotonou (Table 1), especially at night (Sealy et

al., 2003), leading to the lower exposure for drivers at night after aerosol scavenging and the

less driving time in wet season because of the unfavorable weather.

The average PE PM$_{2.5}$ levels are compared to the weekly ambient PM$_{2.5}$ concentrations

(Djossou et al., 2018) obtained in the same area and similar sampling period. The average PE

PM$_{2.5}$ were 3.0 and 2.0 times the ambient values found at DF, and 6.1 and 8.8 times at MT in

dry and wet seasons, respectively. The highest PE PM$_{2.5}$ to ambient (A) (PE/A) ratios were

found at WB, i.e., 10.3 in dry and 10.5 in wet season. Such large PE/A ratios are due to the

impact of waste combustion on the respiratory exposure of residents in this area, especially

on children; on the other hand, high PE/A ratios attribute to the fact that WB site is located in

the poorest region of Abidjan, where the extremely simple stove and wood used at home





(Figure S1d), leading to a very high personal PM$_{2.5}$ exposure level indoors during the cooking

time in this area (especially for student B who was in charge of cooking at home sometimes

recorded in the activity logging and questionnaire). Meanwhile, the ambient PM$_{2.5}$ sampling

equipment at WB was neither fixed very close to (blue marker in Figure 1C) nor in the

downwind direction of the landfill (Djossou et al., 2018), which direct suggests the huge

differences between the ambient and personal exposure PM$_{2.5}$ concentrations.

Moreover, we also compare the daytime PE PM$_{2.5}$ mass concentrations with the daytime

ambient PM$_{2.5}$, which collected in the same area and exactly the same dates as the personal

exposure sampling period. The average daytime women PE PM$_{2.5}$ were 3.7 and 1.2 times of

the ambient PM$_{2.5}$ at DF in dry and wet seasons, respectively, which was similar as the

finding from the comparison with the weekly PM$_{2.5}$ mentioned above. But for the students at

WB and drivers at MT, PE/A ratios were both much smaller than those compared with the

weekly ambient PM$_{2.5}$, 5.1 and 7.0 for the students at WB, 1.9 and 3.3 for the drivers at MT

in dry and wet seasons, respectively. PE/A ratios for students were also showed the highest

values. PE/A ratios observed all above 1.0 and the great variability of PM$_{2.5}$ mass

concentrations between personal exposure and ambient samples imply that fix-point sampling

is likely to underestimate the PM$_{2.5}$ personal exposure and consequent human health hazards,

and confirm the importance of portative PE PM$_{2.5}$ sampling to PM$_{2.5}$ health risk assessment

again.

*3.1.2. PE PM$_{2.5}$ chemical compositions*

Table 4 summarizes the average concentrations of PE PM$_{2.5}$ chemical compositions,

including carbon fractions (OC and EC), water-soluble inorganic ions and several heavy

metals. Total carbon (TC) was the most important chemical species in PE PM$_{2.5}$, accounting

for 24.4%±4.5%, 16.6%±2.0% and 17.8%±4.9% of PE PM$_{2.5}$ mass of women, students and

drivers, respectively. High level of OC proves the strong contribution of combustion sources

to PE PM$_{2.5}$ in SWA in this study (Djossou et al., 2018; Ouafo-Leumbe et al., 2017). The OC

and EC concentrations varied significantly, ranging from 28.3 to 460.0, 8.0 to 189.9 and 14.7

to 65.1 µg m$^{-3}$ for OC and 1.5 to 31.1, 0.8 to 35.1 and 1.9 to 18.2 µg m$^{-3}$ for EC for women,

students and drivers, separately. The OC concentration (83.2 µg m$^{-3}$) and percentage (24.4%)

in women PE PM$_{2.5}$ samples were the highest among the three types of exposed participants,

due to their close contact with the ignition and the direct burning of the solid fuels (wood in

this study) and roasting meat at the workplace, and even cooking at home, etc. However, the

EC concentrations and proportions for these three targets were similar (8.4-10.5 µg m$^{-3}$ and

3.0%-3.5%), meaning that EC is less affected by human activities related to combustion





sources in this study.

The OC and EC ratio (OC/EC) has been used to determine emission and transformation

characteristics of carbonaceous aerosols (Cao et al., 2008). OC/EC averaged 9.9±5.3 for

women at DF, 6.1±0.7 for students at WB, and 5.8±2.7 for drivers at MT. The previous

studies (Cao et al., 2008; Li et al., 2009; Tian et al., 2017) suggested that average OC/EC

characterizes 1.1 as motor vehicle exhaust, 2.7 as coal combustion and 9.0 as biomass

burning. The OC/EC in the present study points out that biomass burning emission was the

main contributor to carbonaceous aerosols for women at DF, and the mixed emissions from

biomass and coal burning, even or/and motor vehicle exhaust dominated the carbonaceous

aerosol sources for students at WB and drivers at MT. The OC/EC was almost always higher

during wet than dry season, which may be related to the fact that the higher relative humidity

in wet season favors the formation of secondary organic carbon (SOC) (Huang et al., 2014).

Drivers' daytime OC/EC shows relatively low (average: 3.7) and stable ratios in wet and dry

seasons, indicating that motor vehicle exhaust was the most important source to drivers' OC

and EC in the daytime, consistent with the working environment of motorcycle drivers in this

study. Personal exposure of women displays the higher (average: 13.9) and more scattered

OC/EC than those collected from students and drivers in wet season (Figure 4), resulting

from the particularly high and dramatic changes individual exposure to obvious carbonaceous

aerosol sources (wood burning and grilling).

In a previous study of Djossou et al. (2018) about OC/EC at ambient (A) environment,

OC/EC in personal exposure samples were about 1.2 and 2.5 times of the ambient values in

dry and wet seasons for women at DF, 1.7 and 2.8 times for students at WB, and 1.1 and 2.0

434 times for drivers at MT. Therefore, the higher OC/EC values in personal exposure samples

resulted from some specific individual's activities and potentially contaminated

microenvironments (Crist et al., 2008; Meng et al., 2009). From the results we can also see

that the influences of precipitation and other meteorological factors on OC/EC of the ambient

samples are less than personal exposure samples (Dry season OC/EC was more comparable

between the ambient and personal exposure samples in this study).

The average concentrations of total measured water soluble inorganic ions were

23.6±12.8, 35.5±18.3 and 22.7±5.0 µg m$^{-3}$ for women at DF, students at WB and drivers at

MT, accounting for 8.5%±1.0%, 12.1%±2.7% and 11.9%±0.4% of PE PM$_{2.5}$ mass,

respectively. Unlike of the ion compositions in polluted cities of China (SO$_4^{2-}$, NO$_3^-$ and

NH$_4^+$ were the most abundant ions in ambient PM$_{2.5}$, accounting for 50%-90% of measured

ions and ~30% of PM$_{2.5}$ mass) (Xu et al., 2016b; Zhang et al., 2013), Ca$^{2+}$, a marker of





fugitive dust, was the most abundant ion, accounting for ~28% (from 25.3% to 29.3%) of

ions in this study, following by $Cl^-$, $SO_4^{2-}$ and $K^+$ for women at DF, $Na^+$ $SO_4^{2-}$ and $Cl^-$ for

students at WB, $SO_4^{2-}$, $Na^+$ and $NO_3^-$ for drivers at MT. It can thus be seen that the particle

resuspension by personal activities is the main contributor to PE $PM_{2.5}$ in SWA (Chen et al.,

2017; Xu et al., 2015). The day and night variations of $Ca^{2+}$ contribution to total ions also

illustrate this conclusion (day=30.6% and night=22.8%). $SO_4^{2-}$ forms primarily through

atmospheric oxidation of $SO_2$ emitted mainly from coal and diesel combustion (Seinfeld and

Pandis, 2006, Xu et al., 2016b). As the second most enriched ion, the average proportion of

$SO_4^{2-}$ was 17.7%, which implies that purification raw coal and diesel (Wang et al., 2013) can

be applied in this area to lead to lower sulfur emissions and therefore decrease the personal

exposure to $SO_4^{2-}$ in $PM_{2.5}$. Drivers' $SO_4^{2-}$ exposure levels were 33% and 40% higher than

women and students respectively, which may indirectly indicate that the emission of $SO_4^{2-}$

precursor $SO_2$ is higher in Cotonou or targeted drivers affected by vehicle emissions are

exposed to high $SO_2$ or $SO_4^{2-}$, especially from the diesel vehicles.

Generally, $Na^+$ and $Cl^-$ were the third and fourth ranked ions. The sampling sites in SWA

cities in this study are all close to the sea and are affected by sea salt particles strongly. It's

also worth noting that biomass burning marker-$K^+$ (Kang et al., 2004; Zhang et al., 2014b)

displays a high absolute average concentration (3.4 μg m$^{-3}$) and percentage (14.5%) in

women PE $PM_{2.5}$ samples, confirming their distinct exposure from biomass burning during

the roasting at the workplace. As we know, $NO_3^-$ derives from $NO_x$ emitted mainly from

motor vehicle exhaust (especially gasoline vehicle), industry and power plants (Seinfeld and

Pandis, 2006; Xu et al., 2016b). Additional consideration here: industry is not well-developed

in this area (much less industry in Cotonou than Abidjan) and is not the main source of $PM_{2.5}$

(Ouafo-Leumbe et al., 2017). It suggests that motor vehicle emission contributed to drivers'

exposure obviously in this study, comparing with women at DF and students at WB,

consistent with the conclusion about $SO_4^{2-}$ above.

The concentrations of 10 targeted heavy metals, including V, Cr, Mn, Co, Ni, Cu, Zn, Sb,

Ba and Pb, can be found in Table 4. The total concentrations of these 10 elements were

1.4±0.3, 3.9±6.5 and 0.8±0.2 μg m$^{-3}$ for women at DF, students at WB and drivers at MT

during the sampling period, accounting for 0.7%±0.4%, 1.0%±1.2% and 0.4%±0.1% of PE

$PM_{2.5}$, correspondingly. The heavy metal exposed concentration of students was 1.8 and 3.9

477  times higher than those for women and drivers, resulting mainly from the garbage

combustion at landfill which emit extremely high level of heavy metals as we known (Wang

et al., 2017b). The D/N ratios ranged from 0.8 to 2.1 for women and drivers, but averaged 4.0



in dry season and 7.0 in wet season for students. There are two reasons for this phenomenon:

the first reason could be related with the intense physical activities from the students and

strong disturbances from landfill workers in the daytime at landfill; the second reason is that

spontaneous combustion of waste occurs frequently during the day, because of less

precipitation and higher air temperature at daytime. Ba, Zn and Mn were found to be the

dominant heavy metals, ~73% of elemental concentration in all samples. It is worth

mentioning that Ba took up a decisive advantage over other elements, accounting for more

than half of all the elements for students. Because Ba is usually added to rubber and plastic

products to improve acid and alkali resistance, at the same time these products are the main

components of the garbage at landfill in this area (Feng et al., 2006). Zn and Mn ranked the

first and second places for drivers at MT, which are mainly derived from the motor oil

additive, tyre wear and brake pads worn (Zhao and Hopke, 2006).

*3.2. Mass balance of personal exposure to $PM_{2.5}$*

Calculation of mass balance for $PM_{2.5}$ is an effective method to figure out the principal

components in $PM_{2.5}$ and for its source discrimination (Gokhale et al., 2008). PE $PM_{2.5}$ mass

in this study can be classified into six parts: organic matter (OM), EC, water-soluble

inorganic ions, geological material (GM), heavy metals and unknown part (Figure 5). The

first five main parts can explain 78.3% to 90.6% of total PE $PM_{2.5}$ mass concentrations in this

study. Unknown part may include water and other undetected substances in PE $PM_{2.5}$. For

OM, since the chemical composition of the aerosol organic fraction is largely unknown,

conversion factor 1.4 (1.4 corrects the organic carbon mass for other constituent associated

with the organic carbon molecule) is generally used (Turpin and Lim, 2001) to shift OC to

OM by equation (5):

$$OM = 1.4 \times OC \qquad (5)$$

based on the equation (5), OM accounted for 34.1%±6.3%, 23.3%±2.8% and 24.9%±6.9% of

PE $PM_{2.5}$ mass for women at DF, students at WB and drivers at MT, respectively, indicating

that there are distinct sources to PE $PM_{2.5}$ OC for women at DF. According to the

questionnaires, the combustion sources, such as roasting meat/peanuts and burning wood,

should be the sources to OC for women personal exposure samples, which consistent with the

results mentioned above.

In addition, Fe has been widely used to estimate the upper limit of GM in previous

studies (Taylor and McLennan, 1985). Fe constitutes about 4.0% of the Earth's crust in dust

of the earth's crust (Cao et al., 2005). The amount of GM is calculated by equation (6):





$$GM = (1/4.0\%) \times Fe \qquad (6)$$

based on this equation, it is found that GM contributed 35.8%±2.1%, 46.0%±3.7% and

42.4%±4.7% of PE $PM_{2.5}$ mass concentrations for women at DF, students at WB and drivers

at MT, respectively. Fugitive dust, including road dust resuspension from disturbance of

motor vehicles and humans, construction dust from uncovered construction sites and the dust

related to burning activities, could be the domination sources to PE $PM_{2.5}$ in this study. OM

and GM show the almost identical proportions (34.1% and 35.8%) of PE $PM_{2.5}$ mass for

women at DF. GM percentages for students and drivers were approximately 10% and 7%

higher than that for women. Therefore, the fugitive dust related contributions are the most

important sources for PE $PM_{2.5}$ in this less developed area, meaning that there are nearly half

PE $PM_{2.5}$ contribution sources of students and drivers attributable to human physical

activities. As mentioned above, it is surprising to note that the secondary formed ions ($SO_4^{2-}$,

$NO_3^-$ and $NH_4^+$), even total measured water-soluble inorganic ions show the exceedingly low

proportions in PE $PM_{2.5}$ for all subjects. This reconfirms the limited contribution to PE $PM_{2.5}$

from secondary formation ionic sources.

From Figure 5, evident diurnal distinguishes are observed in two major chemical

compositions (OM and GM) in this study. We can see that GM exhibits the lower proportion

at night (35.3%) than daytime (47.5%), indicating its close relationship with human activities.

For different seasons, we find the higher GM for each type of participant in dry season,

because of the harmattan haze bringing mineral dust and the lake of precipitation increasing

road dust resuspension. Moreover, OM shows the equal or lower proportion in PE $PM_{2.5}$ at

daytime (25.0%) that nighttime (30.0%), which is mainly related with the meteorological

parameters (they affect the formation of secondary organic carbonaceous aerosol) and

combustion source variations between day and night. There is an exception in the last case,

i.e., OM proportion at daytime women PE $PM_{2.5}$ was much higher (50.8%) than nighttime

(38.2%) in wet season, due to the influence from the damp wood burning at the working time.

As we know, the damp wood burning emits more smoke (PM) than dry wood (Shen et al.,

2013) or change in emission factors (Keita et al., 2018).

*4. Fingerprint organic species in personal exposure to PM$_{2.5}$*

In this section, we use organic fingerprint markers that indicate specific emission

sources to further investigate the sources and detailed characteristics of PE $PM_{2.5}$ for different

populations. Unlike PE $PM_{2.5}$ mass concentration variations (students > women > drivers),

organic fingerprint measured in this study, such as PAHs, PAEs and hopanes (Table 5), show



different concentration orders in this study. The average $PM_{2.5}$-bound PAHs, PAEs and

hopanes mass concentrations were 54.8±20.3, 986.8±82.2 and 27.9±1.0 ng m$^{-3}$ in this study,

respectively, showing a very serious $PM_{2.5}$ organic pollution in SWA region. The descending

following orders were women > students > drivers for PAHs, students > women > drivers for

PAEs, and drivers > women > students for hopanes (Table 5 and Figure 6).

*4.1. PAHs*

The total quantified PAHs (ΣPAHs) accounted for 0.12‰-0.21‰ of PE $PM_{2.5}$ mass

concentration. BbF was the most abundant PAH for women at DF, followed by BaP and IcdP.

The average BbF concentration (the maker of low temperature combustion, such as wood

burning) was 11.6±19.2 ng m$^{-3}$, accounting for up to approximately 15.0% of the ΣPAHs for

women PE samples (Table 5) (Wang et al., 2006). While the most abundant PAHs for

students at WB and drivers at MT were IcdP (6.4±4.5 ng m$^{-3}$) and BghiP (6.4±0.5 ng m$^{-3}$),

respectively, which indicate the contributions from the waste incineration or high temperature

fuel combustion (gasoline vehicle emission) (Baek et al., 1991; Wang et al., 2006). The

ΣPAHs average concentrations in wet season increased 326% and 52% for women at DF

(125.4±54.8 ng m$^{-3}$) and drivers at MT (44.6±10.8 ng m$^{-3}$) than those in dry season (29.4±5.6

and 29.4±4.4 ng m$^{-3}$ respectively), while the ΣPAHs decreased 42% in wet season (36.8±15.7

565      ng m$^{-3}$) compared with dry season (62.9±45.0 ng m$^{-3}$) for students at WB. The dramatic

increase in women's exposure to PAHs is mainly due to the increase in humidity (moisture

content) of the wood used for grilling meat in wet season, resulting in PAH emissions sharp

raising from wood combustion (Shen et al., 2013). The restraint of waste combustion in wet

season is the main factor in the decrease of students' exposure to $PM_{2.5}$-bound PAHs at

landfill, in accordance with PE $PM_{2.5}$ mass seasonal change pattern. PE PAHs concentrations

were measured in Cotonou in the previous study (Fanou et al., 2006), the result showed that

the level of total PAHs associated with particles ranged from 76.21 to 103.23 ng m$^{-3}$ for 35

taxi-moto drivers in March 2001. The PAH levels determined in this study for drivers at MT

site was 50%-64% lower than the values in Fanou et al. (2006) study, suggesting that the

motorbike driver exposure to PAHs in this region has improved.

As shown in Figure 6A, differing from the almost unchanged diurnal variation (daytime >

nighttime) of PE $PM_{2.5}$ and its major chemical components discussed above, PE PAHs show

unstable diurnal variations for these three types of target populations: 1) Women at DF: the

daytime concentrations during the wet and dry seasons were both higher than those at

nighttime, suggesting that women's intensive combustion activities at daytime (roasting meat

and burning wood) strongly impacted the PE PAHs. The average D/N ratios were 1.7 in dry



season with the 12-hour average ΣPAHs of 37.4±25.1 ng m$^{-3}$ for daytime and 21.4±17.2 ng

m$^{-3}$ for nighttime and 3.5 in wet season with 195.6±121.9 ng m$^{-3}$ for daytime and 55.3±44.3

584    ng m$^{-3}$ for nighttime; 2) Students at WB: nighttime PE PAHs were higher in dry season and

lower in wet season compared with daytime levels, with the average D/N ratios of 0.7 and 1.8,

respectively. The higher concentrations of combustion markers-BbF and BeP were observed

during the day, while the higher concentrations of gasoline vehicle emission markers-DahA

and BghiP were found at night (Baek et al., 1991; Wang et al., 2006), which was related to

the garbage truck for waste transportation from city to the landfill during night. Moreover, we

should also note that the impact of garbage truck emission was offset by PM$_{2.5}$ wet deposition

during the wet season; 3) Drivers at MT: we are surprised to see that the average dry season

D/N ratio was 0.8 with the ΣPAHs of 26.3±7.6 ng m$^{-3}$ for daytime and 32.5±13.8 ng m$^{-3}$ for

nighttime, and the average wet season D/N ratio was 0.3 with 21.9±8.4 ng m$^{-3}$ for daytime

and 67.3±23.7 ng m$^{-3}$ for nighttime, respectively. The high nighttime ΣPAHs concentrations

and low D/N ratios for drivers in this study may be explained by the possibility that there are

potential combustion sources (PAH sources) around participant drivers (especially around

drivers' home at night) in Cotonou, Benin at night rather than the motor vehicle exhaust,

especially in wet season (combustion emission marker BaP was the highest PAH species at

night in wet season), although the drivers exposed to the traffic emissions during the night

working time (18:30 to 21:00 UTC). Further studies are required to confirm the findings and

figure out the reasons. Even so, the highest PAH individual species for drivers PE samples

was BghiP (gasoline vehicle emission marker) in both wet and dry seasons, proving the

obvious influence from the motor vehicle emissions to motorcycle drivers (Baek et al., 1991;

Wang et al., 2006).

Diagnostic ratios of PAHs with similar molecular weights have been widely used in

source identification (Tobiszewski and Namiesnik, 2012; Yunker et al., 2002). In our study,

the average values of BeP/(BeP+BaP) and IcdP/(IcdP+BghiP) were 0.47 and 0.52 for women

at DF, 0.51 and 0.52 for students at WB, and 0.64 and 0.34 for drivers at MT, respectively

(Figure 7), showing that the impacts of different atmospheric pollution sources on the

different type participants are very significant, and that the diagnostic ratios of PAHs can be

applied to identify the source of PAHs in PE PM$_{2.5}$ effectively. The average BeP/(BeP+BaP)

ratio ranged from 0.47 to 0.64, comparable with those reported in Guangzhou (0.41-0.72) and

Xi'an (0.59-0.73) of China (Li et al., 2005; Xu et al., 2018b) and lower than that reported in

Shanghai (all samples > 0.70), China (Feng et al., 2006), implying the low oxidability of the

PAHs in SWA cities (less developed than Chinese cities). PAHs in drivers' PE samples are




more prone to aging (the average ratio was 1.3-1.4 times of those for women and students)

because of the re-suspension of road dusts where PAHs are attached to (longer residence

lifetime) and longer outdoor activity time (more sunlight); and more fine and ultra-fine

particles-bound PAHs from high-temperature combustion in motor vehicular engine, which

are more easily photochemical oxidation in the air (Baek et al., 1991; Lima et al., 2005). The

difference of BeP/(BeP+BaP) ratios in dry and wet seasons is not obvious and no fixed rule.

However, this ratio exhibits a significant day-night change, with the values of 0.59 at daytime

and 0.49 at nighttime. It means that more beneficial meteorological conditions at daytime

(such as more sunlight) and stronger individual physical activity (increasing the time of

particulate re-suspension) are more conducive to the aging of $PM_{2.5}$ and its bounded PAHs.

Moreover, IcdP/(BghiP+IcdP) of < 0.2, 0.2-0.5 and > 0.5 represent petrogenic, petroleum

combustion and a mix of grass, wood, and coal combustions, respectively (Yunker et al.,

2002). The quite low ratios for drivers at MT (0.34) indicates that the PAHs in those samples

were mainly produced from motor vehicle emissions (petroleum combustion), while grass,

wood and coal combustions were more dominant for women at DF (0.52) and students at WB

(0.52) (Figure 7). IcdP/(IcdP+BghiP) ratio in all samples from our study shows not

significant seasonal variation.

*4.2. Phthalate esters (PAEs)*

Phthalate esters are widely used plasticizers in plastic materials and can be released into

the air from the matrix evaporation and plastics combustion (Gu et al., 2010; Wang et al.,

2017a). The personal exposure levels of PAEs could be mainly attributed to the usage of the

household products, painting material at home, plastic waste incineration and municipal

sewage release (Zhang et al., 2014a). The total concentrations of six phthalate esters (DMP,

DEP, DBP, BBP, DEHP and DNOP) and one plasticizer (DEHA) (named ΣPAEs for all these

seven species) were 882.0±193.3, 1380.4±335.2 and 698.1±192.4 ng m$^{-3}$, respectively, for

women at DF, students at WB and drivers at MT (Table 5). DEHP was the most dominant

PAE species, followed by DBP in this study for all the three kinds of participants. DEHP is

mainly used as a plasticizer for polyvinyl chloride (PVC). And together with DBP, they are

the most widely used phthalate esters globally (Meng et al., 2014). The average DEHP and

DBP concentrations were 543.6 and 304.6 ng m$^{-3}$, accounting for up to approximately 55.1%

and 30.9% of the ΣPAEs, respectively (Figure 6B). The elevated ΣPAEs for students at WB

in this study are mostly result from combustion of the plastic products at landfill. The results

in this study are similar as the previous studies carried out in Xi'an, Tianjin of China (Kong





et al., 2013; Wang et al., 2017a). The ΣPAEs ranged from 376.6 to 1074 ng m$^{-3}$ in outdoors,

and from 469.2 to 1537 ng m$^{-3}$ in student classrooms in Xi'an (Wang et al., 2017a), in which

DEHP and DBP were also the dominant species, totally accounting for 68% and 73% of the

ΣPAEs in outdoor and indoor environments, respectively.

The average concentrations of the ΣPAEs for women at DF, students at WB and drivers

at MT were comparable in dry season. But the average concentrations were 927.2±154.9,

1929.8±340.4 and 594.6±16.6 ng m$^{-3}$ in wet season in this study, 1.1, 2.3 and 0.7 times of the

ΣPAEs values in dry season (Figure 6B). A significant increase in students PE ΣPAEs at WB

in wet season can be attributed to the enhanced PAEs emission in the day (3173.6±1028.3 ng

m$^{-3}$), consistent with the findings on PE PM$_{2.5}$ above. Dry and wet seasons led to almost

similar PAEs profiles with different day and night variations (Figure 6B). The average D/N

ratios of the ΣPAEs in dry season show limited changes, with the values of 1.0, 1.0 and 1.3,

respectively, for women, students and drivers; while 1.1, 4.6 and 0.7 in wet season.

Noticeably different D/N ratios between two seasons observed in this study for students at

WB is interrelated with the human activities (specially related to the plastic material

emissions) and the subdued waste spontaneous combustion resulting from diurnal variations

of meteorological parameters (more precipitation at night in wet season) mentioned in Sect.

3.1.1.

*4.3. Hopanes*

Hopanes have been used as markers for fossil fuel combustion, especially for petroleum

combustion (Simoneit, 1999; Wang et al., 2009). The average concentration of drivers who

exposed to the eight hopanes (Σhopanes) in this study was 50.9±7.9 ng m$^{-3}$, 2.0 and 2.3 times

higher than for women at DF (17.1±6.4 ng m$^{-3}$) and students at WB (15.6±6.1 ng m$^{-3}$) (Table

5), respectively, which proves the extremely high driver personal respiratory exposure

contribution from the motor vehicle emissions (gasoline combustion) in this study. Then, it is

important to note that numbers of automobiles are rapidly increasing in SWA cities, which

further exacerbates the air pollution and related health problems there. The Σhopanes show

the unobvious seasonal variations for three kinds of exposure participants, i.e., 0.9, 1.8 and

0.7 times Σhopane concentrations were observed in dry season of those in wet season.

Although the Σhopane concentrations were changeable in this study among three sites, the

distribution of individual species of hopanes were similar for each participant. αβ-NH and

αβ-HH were two dominant hopanes in all PE PM$_{2.5}$ samples, with the average concentrations



of 6.0 and 6.5 ng m$^{-3}$ and the percentages of 21.4% and 23.3% of the Σhopanes, respectively

(Table 5 and Figure 6C).

Compared with D/N ratios of the ΣPAHs and ΣPAEs, hopanes exhibit a more stable

diurnal trend in this study, namely, daytime concentrations were always greater than

nighttime, owing to the obvious traffic emissions during the day. For women at DF, D/N ratio

was both 2.0 in dry and wet seasons, with the Σhopanes of 24.0±11.1 and 12.2±5.0 ng m$^{-3}$ for

daytime and nighttime in dry season, and 21.4±17.5 and 10.9±3.6 ng m$^{-3}$ in wet season.

Emphasize that D/N ratio of the Σhopane for drivers at MT presents the highest value (11.5)

for all the detected chemical species in this study, with the concentrations of 78.0±19.1 and

44.9±16.4 ng m$^{-3}$ for daytime and nighttime in dry season, and 74.2±16.3 and 6.5±1.7 ng m$^{-3}$

in wet season. We notice that the daytime concentrations were comparable for drivers

between two seasons, while the nighttime hopanes in wet season were washed away by

rainfall mostly, resulting in a very large drop in concentration levels.

Therefore, although PAHs, PAEs and hopanes are not abundant components in PE PM$_{2.5}$,

these fingerprint organics can more accurately trace the contribution of air pollution sources

to PM$_{2.5}$. The PAHs, PAEs and hopanes representing emissions from combustion sources,

plastics emissions and fossil fuel combustion emissions (gasoline vehicles) respectively are

very well matched to the potential air pollution sources around these three type participants in

this study. The results not only indicate that the PM$_{2.5}$ respiratory exposure was strongly

contributed from the environmental pollution sources and individual activities, but also prove

the successful application of organic tracers in human exposure study.

*5. Health risk assessment of personal exposure to PM$_{2.5}$*

Non-cancer risks of four heavy metals (Mn, Ni, Zn and Pb) and cancer risks of PAHs

and PAEs via inhalation exposure way for women at DF, students at WB and drivers at MT

are shown in Table 6. In general, the non-carcinogenic risks of Mn and Pb were relatively

higher than those of Ni and Zn, but still well-behind the international threshold value (1.0).

Among those four metals, Hazard Quotient (HQ) of Pb in wet season for students at WB was

the highest (2.95E-02), which suggests that Pb non-carcinogenic risk to children is obvious in

that area compared with other participants and metals. Except that Ni shows the stable wet

season greater non-carcinogenic risk than dry season for all three kinds targets, there was no

stable change in dry/wet season risks of other components. Summing up these four metals,

Hazard Index (HI) values for women at DF, students at WB and drivers at MT in dry and wet

seasons were also represented in Table 6. Dry/wet season ratios of HI were 0.9, 0.5 and 2.3



for women, students and drivers, suggesting that the non-cancer risk of personal exposure to

metals in PM$_{2.5}$ in dry season was much higher than that in wet season for drivers, owing to a

mass of fugitive dust on the road in dry season. Moreover, the yearly average HI levels were

8.06E-03, 4.13E-02 and 8.68E-03 for women at DF, students at WB and drivers at MT,

respectively, showing the highest non-cancer health risks from the heavy metals in PM$_{2.5}$ for

students, 5.1 and 4.8 times of women and drivers. Overall, Mn, Zn, Ni, Pb and HI were all

within the safety limit for all populations involved in this study, pointing out the negligible

non-cancer health risks of heavy metals in PM$_{2.5}$ in SWA region.

In Table 6, the ILCRs of PAHs were all beyond $1\times10^{-6}$ (international acceptable level),

suggesting non-negligible cancer risks of PAHs for women at DF, students at WB and drivers

at MT whenever dry or wet season. Meanwhile, the ILCRs of PAEs were all below $1\times10^{-6}$,

well within the safety limit of cancer risk. For all types of targets, PE PM$_{2.5}$-bound PAHs and

PAEs in wet season were more likely to cause cancer risks than dry season; thus, the seasonal

changes, mainly due to increased humidity, result in an increased personal exposure cancer

risks to toxic organic species in PM$_{2.5}$. In dry season, the average ILCR values of PAHs were

comparable for women and drivers, both ~50% lower than those for students, implying the

high toxicity originated from the waste burning sources and high sensitivity to juveniles. In

wet season, PAHs exhibit the highest ILCR for women at DF, 2.5 and 2.7 times of those for

students and drivers, respectively. It can be seen that the domestic wood burning and grilling

meat can trigger nearly ten times safe limit of cancer risks to target women in this study. The

cancer risks of PAEs show the similar pattern in dry and wet seasons, with the descending

order of students at WB > women at DF > drivers at MT (Yang et al., 2011). The

carcinogenic risks of PAEs for drivers in traffic environment was the lowest, much lower (45%

and 76% lower in dry and wet seasons) than PAEs for students who are close to the source of

waste incineration. In a word, the ILCRs of PAHs exceeded the threshold value of $1\times10^{-6}$ for

all the participants, indicating that the carcinogenic PAHs are a threat to the individual's

health and subsequently alerting a need of effective control in SWA. However, PAEs show

limited carcinogenic risks in this study, but the effect of waste burning source to students is

needed to pay more attention and reasonable controls for both PM$_{2.5}$-bound heavy metals and

organic compounds.

In addition, it should be noted that the non-cancer and cancer risks could be potentially

underestimated since many toxic chemical components could not be detected in this study. It

is concluded based on the data that different targets present different levels of risks from

different chemical species in PE PM$_{2.5}$ from various air pollution sources. We must pay





attention to heavy metal non-cancer health risks via inhalation way, especially Pb and Mn for

students at WB site as well as PAHs cancer risks for women at DF site in wet season in SWA

region.

**6. Conclusions**

We explore the chemical characteristics and health risks of personal exposure to $PM_{2.5}$

(PE $PM_{2.5}$) from different typical anthropogenic air pollution sources in Southern West Africa.

Our study finds that organic matter and geological material are the almost identical

proportions (34.1% and 35.8%) for women at domestic fire site. Nearly half contribution to

PE $PM_{2.5}$ for students at waste burning site and drivers at motorcycle traffic site comes from

fugitive dust. Therefore, the primary source (mainly dust) is the most important source for PE

$PM_{2.5}$ in these undeveloped regions. The contribution to PE $PM_{2.5}$ from heavy metals was

higher for students (1.0%), owing to the waste burning emissions strongly, leading to the

highest non-cancer risk among these three kinds of participants, as well as the extremely high

PAEs concentrations (indicator of plastic emissions). PE $PM_{2.5}$-bound PAHs concentration for

women at domestic fire site was 1.6 times for students and 2.1 times for drivers, which is

mainly attributed to the wood burning and grilling meat activities, resulting in approximately

five times higher of international cancer risk safe limit (nearly ten time of threshold value in

wet season). Drivers' exposure to hopanes in PE $PM_{2.5}$ was 2.0-2.3 times higher than women

and students, correlating with the elevated traffic emissions on road environment well.

This work can be regarded as the first attempt in underdeveloped country of Africa at the

current condition, although there are some drawbacks, such as relatively short sampling

period and a limited number of participants. More investigations on personal exposure and

related potential health effects by cohort study method will be considered in the further. The

policy implication of our findings is that developing and implementing appropriate

preventive and control measures on different $PM_{2.5}$ anthropogenic sources in different regions

are appropriate, such as using dry wood for barbecues for the female workers and improving

waste treatment equipment at landfill as soon as possible to reduce waste inorganized stack

and open combustion.

**Acknowledgments**

This study was supported by the Natural Science Foundation of China (NSFC)

(41503096) and has received funding from the European Union 7[th] Framework Programme

(FP7/2007-2013) under Grant Agreement no. 603502 (EU project DACCIWA: Dynamics-





Aerosol-Chemistry-Cloud Interactions in West Africa). Supports from open fund by Jiangsu

Key Laboratory of Atmospheric Environment Monitoring and Pollution Control (KHK1712),

and a Project Funded by the Priority Academic Program Development of Jiangsu Higher

Education Institutions (PAPD), and open fund by State Key Laboratory of Loess and

Quaternary Geology, Institute of Earth Environment, CAS (SKLLQG1722) are also thanked.

In addition, the authors thank all the personal exposure sampling participants who helped a

lot in this study.

**Author Contributions**

H.X. and C.L. conceived and designed the study. H.X., J.-F.L., C.L. and B.G.

contributed to the literature search, data analysis/interpretation and manuscript writing. J.-F.L.,

C.L., B.G., V.Y., A.A., K.H., S.H., Z.S. and J.C. contributed to manuscript revision. H.X., J.-

F.L., E.G., J.A and L.L. carried out the particulate samples collection and chemical

experiments, analyzed the experimental data.

**Additional Information**

Fig. S1 accompany this manuscript can be found in Supplementary Information.

**Competing financial interests**

The authors declare no competing financial interests.

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



**Figure Caption:**

**Figure 1.** Locations of the sampling sites within the cities. A: Domestic Fires (DF) site at the Yopougon-Lubafrique market in Abidjan; B: Waste Burning (WB) site at the landfill of Akeoudo in Abidjan, the location of the long-term sampling site is given by the blue marker; and C: Motorcycle Traffic (MT) site at Dantokpa area in Cotonou.

**Figure 2**. Pictures showing the sampling sites and corresponding participants: (a) women at DF; (b) students at WB; (c) drivers at MT.

**Figure 3.** Personal exposure to $PM_{2.5}$ mass concentrations of woman at DF, student at WB and driver at MT in dry season (January) and wet season (July) of 2016 in SWA area.

**Figure 4** Variations of OC/EC ratios in personal exposure to $PM_{2.5}$ samples for women at DF, students at WB and drivers at MT (The box plots indicate the average concentration and the min, $1^{st}$, $25^{th}$, $50^{th}$, $75^{th}$, $99^{th}$ and max percentiles).

**Figure 5.** Personal exposure to $PM_{2.5}$ mass concentration closures for women at DF, students at WB and drivers at MT in different sampling seasons.

**Figure 6.** Distributions of A: PAHs; B: PAEs; and C: hopanes in $PM_{2.5}$ personal exposure samples for women at DF, students at WB and drivers at MT in dry and wet seasons of 2016.

**Figure 7.** Correlations between PAHs diagnostic ratios (average ratio points of each type participant indicate day and night value respectively).



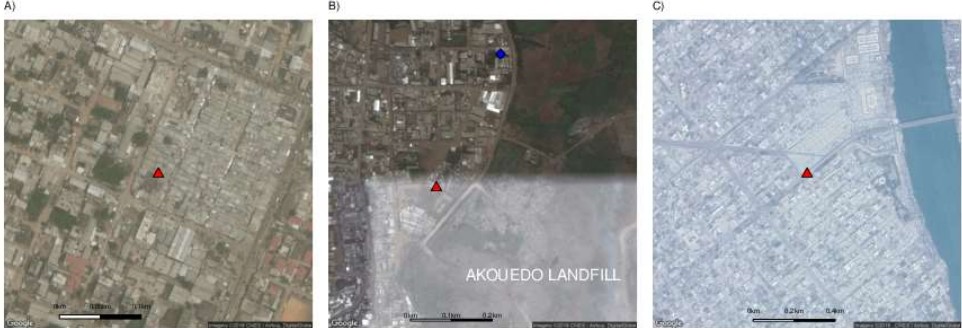

**Figure 1.**



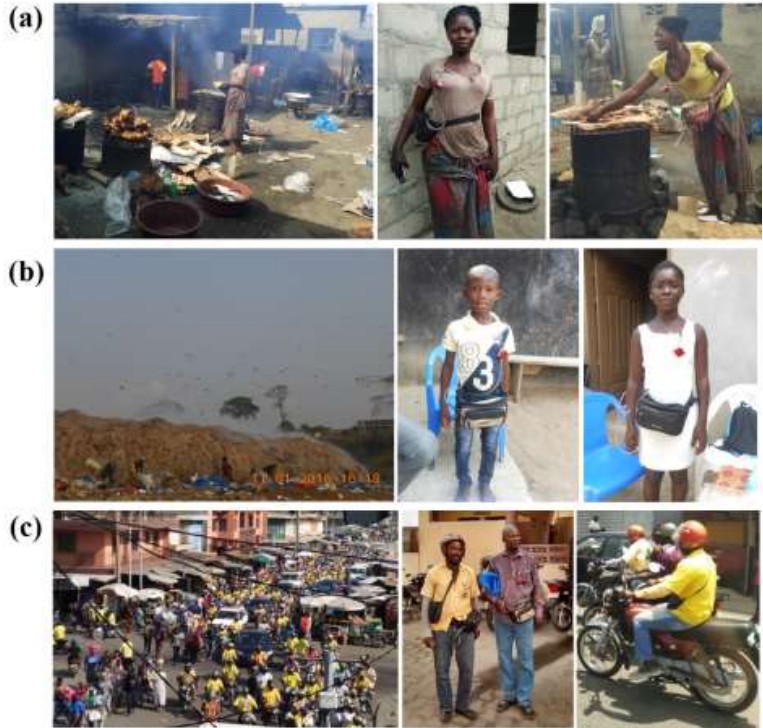

**Figure 2**.





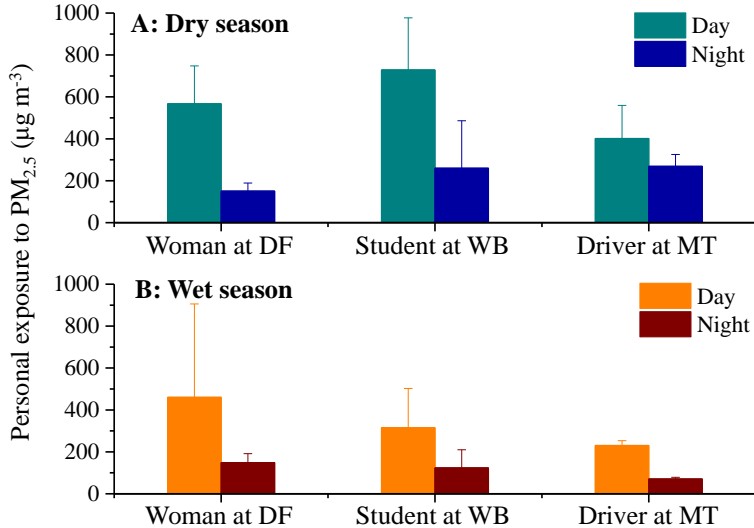

**Figure 3.**





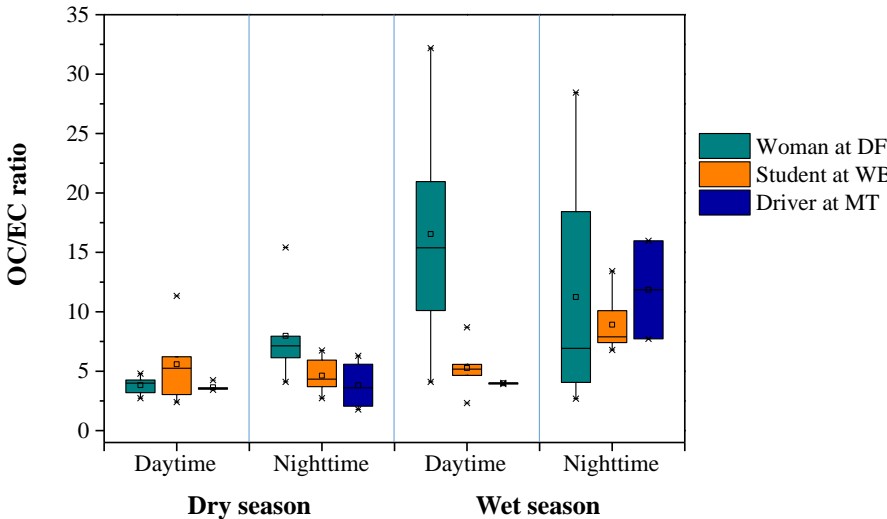

Figure 4.




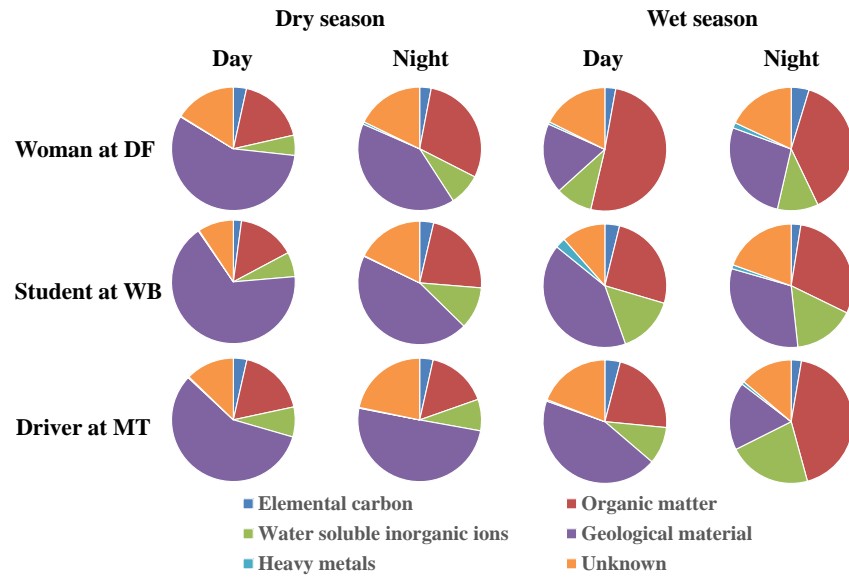

**Figure 5.**

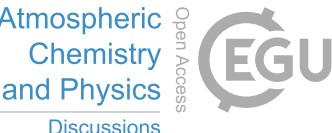





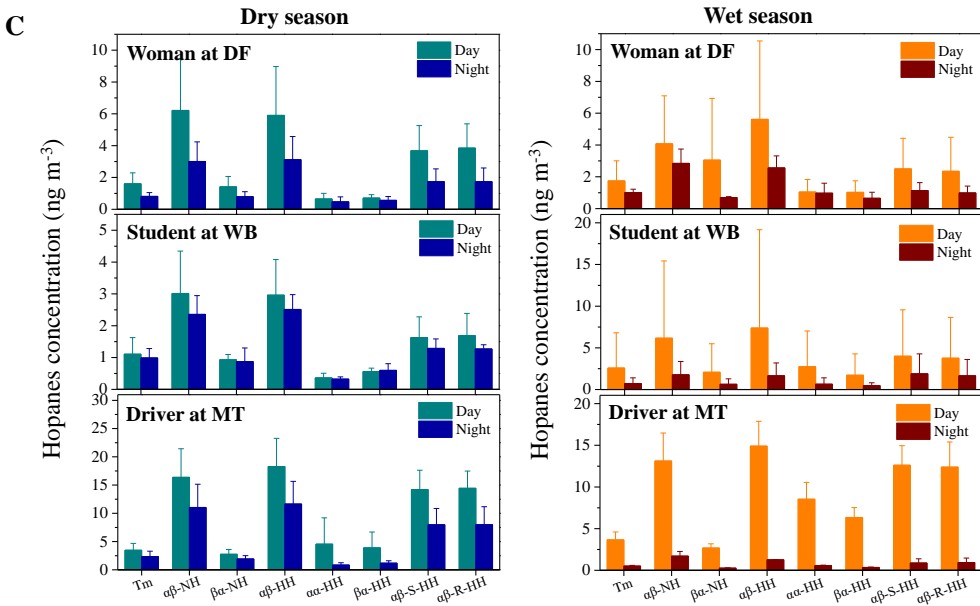

**Figure 6.**





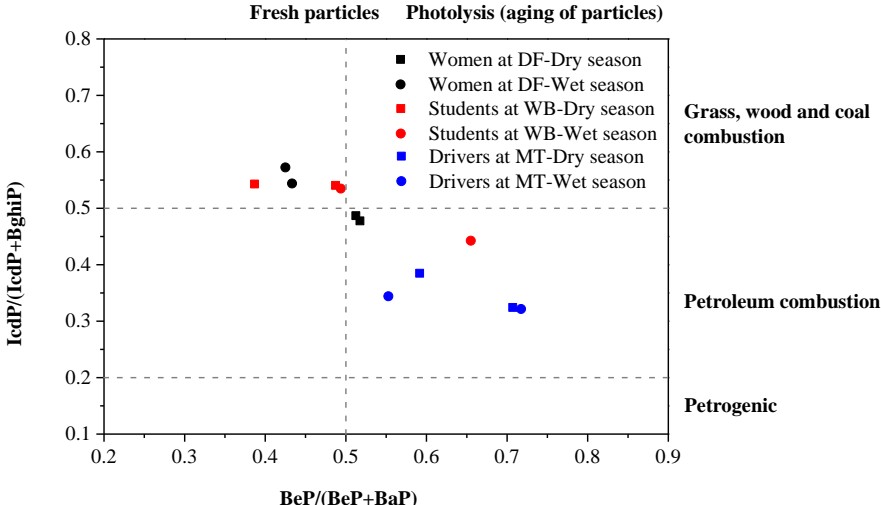

**Figure 7.**





**Table 1** Meteorological parameters of the studied two cities during the dry (December 2015

to March 2016) and wet (April to July 2016) seasons.

| | Season | Abidjan | Cotonou |
|---|---|---|---|
| Mean daily air temperature (°C) | Dry | 28.0 | 28.3 |
| | Wet | 27.5 | 27.7 |
| Total rainfall (mm) | Dry | 268 | 92 |
| | Wet | 626 | 558 |
| Mean wind speed (m s$^{-1}$) | Dry | 3.0 | 3.0 |
| | Wet | 3.4 | 4.3 |





**Table 2** Definitions and recommended values of the parameters in equations (1-4) in this

study.

| Parameter | Definition (unit) | Value used in this study (reference) |
|---|---|---|
| D | average daily exposure dose (mg kg$^{-1}$ day$^{-1}$) | / |
| C | heavy metals concentrations in equations (ng m$^{-3}$) | / |
| R | inhalation rate, air volume a child inhaled each day (m$^3$ day$^{-1}$) | 16.0 for women and drivers; 15.2 for students (USEPA, 2011) |
| EF | exposure frequency (day year$^{-1}$) | 130 for women and drivers (half working days); 182 for students (half year) |
| ED | exposure duration (year) | 30 for women and drivers (working years); 15 for students (before going to high school) |
| BW | body weight (kg) | 62.5 for women[a]; 37.5 for students[a]; 85.0 for drivers[a] |
| AT | averaging time (day) | 30 or 15 × 365 (non-cancer); 70 × 365 (cancer) |
| *cf* | conversion factor (kg mg$^{-1}$) | 10$^{-6}$ |
| HQ | hazard quotient | / |
| RfD | reference dose, estimated as the maximum permissible risk on human by daily exposure (mg kg$^{-1}$ day$^{-1}$) | Table 3 |
| HI | hazard index | / |
| ILCR | incremental lifetime cancer risk (ILCR) | / |
| CSF | cancer slope factor (mg kg$^{-1}$ day$^{-1}$)$^{-1}$ | Table 3 |
| [BaP]$_{eq}$ | equivalent BaP toxicity concentration (ng m$^{-3}$) | / |
| C$_i$ | individual PAH species concentration (ng m$^{-3}$) (i means target PAH species) | / |
| TEF$_i$ | toxicity equivalency factor of each target PAH compound (i means target PAH species) | (Nisbet and Lagoy, 1992) |

a: Measured in this study.





**Table 3** Reference dose (RfD) (mg kg$^{-1}$ day$^{-1}$) and cancer slope factor (CSF) (mg kg$^{-1}$ day$^{-1}$)$^{-1}$

via inhalation exposure way used in this study.

|      | RfD | CSF | Reference |
|------|-----|-----|-----------|
| Mn   | $1.8\times10^{-3}$ | / | Liu et al., 2015 |
| Ni   | $5.4\times10^{-3}$ | / | Zhou et al., 2014; Liu et al., 2015 |
| Zn   | $3.0\times10^{-1}$ | / | Zhou et al., 2014 |
| Pb   | $3.5\times10^{-3}$ | / | Zhou et al., 2014; Hu et al., 2012 |
| BaP  | / | 3.140 | USEPA, 2011 |
| DEHP | / | 0.014 | USEPA, 1997; Wang et al., 2017a |





**Table 4** Statistical analysis (arithmetic mean±standard deviation) of personal exposure to $PM_{2.5}$ mass concentrations and the chemical
compositions (units: µg m$^{-3}$) during the sampling period in SWA region.

| | Dry season | | | | | | Wet season | | | | | |
|---|---|---|---|---|---|---|---|---|---|---|---|---|
| | Women at DF | | Students at WB | | Drivers at MT | | Women at DF | | Students at WB | | Drivers at MT | |
| | Daytime | Nighttime | Daytime | Nighttime | Daytime | Nighttime | Daytime | Nighttime | Daytime | Nighttime | Daytime | Nighttime |
| PE $PM_{2.5}$ | 567.0±180.6 | 150.6±38.5 | 728.5±248.5 | 260±226.1 | 401.3±158.0 | 269.0±56.1 | 460.5±445.2 | 148.6±42.9 | 315.2±186.9 | 123.7±86.1 | 230.4±22.8 | 70.7±8.1 |
| OC | 72.4±24.6 | 31±5.0 | 85.0±57.4 | 40.9±34.4 | 49.5±12.5 | 31.8±14.2 | 189.3±197.8 | 40.1±9.3 | 65.2±65.2 | 28.5±26.8 | 37.0±3.5 | 22.2±10.6 |
| EC | 19.5±7.3 | 4.7±2.2 | 15.0±4.7 | 8.6±5.7 | 13.6±3.6 | 9.0±2.3 | 11.5±10.8 | 6.3±3.7 | 12.3±11.4 | 3.6±3.6 | 9.3±0.8 | 1.9±0.0 |
| Total carbon | 91.9±31.1 | 35.7±6.8 | 100.0±60.1 | 49.5±39.5 | 63.1±16.0 | 40.8±13.6 | 200.8±207.1 | 46.3±7.2 | 77.4±76.2 | 32.1±30.3 | 46.3±4.2 | 24.1±10.6 |
| $Cl^-$ | 4.4±1.3 | 1.6±0.6 | 6.5±3.6 | 6.4±9.4 | 2.4±0.8 | 2.2±0.6 | 8.6±8.4 | 1.9±1.0 | 4.6±5.4 | 1.9±0.7 | 3.1±0.2 | 2.3±0.2 |
| $NO_3^-$ | 2.7±0.7 | 2.2±1.4 | 5.5±1.3 | 3.0±0.7 | 3.7±1.3 | 2.7±0.5 | 2.2±0.8 | 1.6±0.7 | 5.0±6.0 | 1.8±1.3 | 1.6±0.2 | 1.2±0.1 |
| $SO_4^{2-}$ | 4.0±1.1 | 1.8±0.6 | 7.5±2.5 | 3.6±0.9 | 7.5±2.5 | 5.3±0.6 | 6.8±5.2 | 2.3±0.8 | 6.4±5.9 | 2.3±0.4 | 5.2±0.3 | 3.2±0.5 |
| $Na^+$ | 2.9±0.4 | 1.6±0.3 | 4.1±1.1 | 1.9±0.8 | 3.3±1.1 | 2.4±0.3 | 4.2±2.2 | 4.4±1.7 | 16.2±17.3 | 3.3±3.1 | 3.6±0.2 | 2.6±0.1 |
| $NH_4^+$ | 0.6±0.2 | 0.4±0.5 | 1.4±0.4 | 3.0±4.1 | 1.1±0.2 | 0.9±0.2 | 0.6±0.5 | 0.1±0.0 | 0.6±0.2 | 0.4±0.3 | 0.7±0.0 | 0.1±0.0 |
| $K^+$ | 3.2±0.6 | 1.7±0.6 | 5.8±4.0 | 2.2±0.8 | 1.9±0.4 | 2.1±0.9 | 7.6±8.0 | 1.3±0.8 | 3.3±4.4 | 1.3±0.6 | 1.1±0.0 | 3.6±1.5 |
| $Mg^{2+}$ | 0.6±0.2 | 0.2±0.1 | 0.8±0.3 | 0.3±0.2 | 0.4±0.2 | 0.3±0.1 | 1.1±1.2 | 0.3±0.1 | 1.0±0.9 | 0.3±0.2 | 0.3±0.0 | 0.2±0.0 |
| $Ca^{2+}$ | 11.0±3.2 | 3.1±0.9 | 14.9±4.5 | 4.9±3.2 | 10.6±5.5 | 6.0±1.2 | 6.6±4.3 | 3.2±0.8 | 17.3±13.9 | 4.5±3.8 | 6.8±0.3 | 2.3±0.1 |
| Total ions | 29.3±6.6 | 12.5±3.7 | 46.6±15.4 | 25.2±18.8 | 30.9±11.9 | 21.9±3.2 | 37.6±29.5 | 15.1±2.2 | 54.4±50.0 | 15.8±8.8 | 22.3±1.0 | 15.5±1.9 |
| Fe | 14.61±5.25 | 2.64±0.36 | 21.17±4.64 | 4.85±3.30 | 10.99±6.50 | 5.90±0.37 | 3.37±3.34 | 1.87±0.96 | 5.07±1.74 | 1.76±1.24 | 4.56±0.64 | 0.57±0.05 |
| V | 0.04±0.02 | 0.00±0.00 | 0.07±0.02 | 0.02±0.01 | 0.03±0.02 | 0.01±0.01 | 0.01±0.01 | 0.00±0.00 | 0.03±0.03 | 0.01±0.01 | 0.01±0.00 | 0.01±0.00 |
| Cr | 0.04±0.02 | 0.01±0.00 | 0.06±0.02 | 0.01±0.01 | 0.03±0.03 | 0.01±0.01 | 0.05±0.02 | 0.06±0.03 | 0.31±0.35 | 0.04±0.05 | 0.03±0.00 | 0.03±0.00 |
| Mn | 0.18±0.06 | 0.04±0.03 | 0.29±0.08 | 0.07±0.04 | 0.35±0.12 | 0.21±0.11 | 0.14±0.16 | 0.04±0.00 | 0.37±0.36 | 0.06±0.06 | 0.17±0.02 | 0.04±0.00 |
| Co | 0.05±0.02 | 0.01±0.01 | 0.09±0.02 | 0.01±0.01 | 0.05±0.03 | 0.02±0.02 | 0.02±0.02 | 0.02±0.02 | 0.04±0.05 | 0.02±0.02 | 0.02±0.01 | 0.01±0.00 |
| Ni | 0.02±0.01 | 0.00±0.00 | 0.02±0.01 | 0.01±0.01 | 0.02±0.01 | 0.01±0.01 | 0.02±0.02 | 0.03±0.02 | 0.12±0.14 | 0.02±0.03 | 0.02±0.00 | 0.01±0.00 |
| Cu | 0.04±0.01 | 0.02±0.01 | 0.14±0.03 | 0.02±0.01 | 0.05±0.03 | 0.03±0.01 | 0.13±0.07 | 0.13±0.07 | 0.67±0.81 | 0.10±0.09 | 0.07±0.02 | 0.06±0.01 |
| Zn | 0.40±0.22 | 0.55±0.73 | 0.49±0.19 | 0.15±0.12 | 0.33±0.16 | 0.19±0.07 | 0.51±0.32 | 0.32±0.17 | 1.41±1.55 | 0.26±0.27 | 0.29±0.04 | 0.12±0.00 |



| | | | | | | | | | | | | |
|---|---|---|---|---|---|---|---|---|---|---|---|---|
| Sb | 0.02±0.01 | 0.05±0.02 | 0.02±0.02 | 0.00±0.00 | 0.02±0.04 | 0.01±0.01 | 0.12±0.08 | 0.21±0.18 | 1.16±1.38 | 0.22±0.29 | 0.07±0.04 | 0.08±0.09 |
| Ba | 0.19±0.09 | 0.16±0.12 | 0.25±0.11 | 0.07±0.09 | 0.22±0.18 | 0.05±0.07 | 0.47±0.39 | 1.02±0.60 | 6.80±8.30 | 0.84±1.41 | 0.18±0.18 | 0.14±0.01 |
| Pb | 0.07±0.03 | 0.07±0.07 | 0.17±0.07 | 0.04±0.03 | 0.07±0.05 | 0.02±0.03 | 0.14±0.02 | 0.09±0.03 | 0.92±1.01 | 0.13±0.18 | 0.05±0.02 | 0.03±0.01 |
| **Heavy metals** | **1.05±0.28** | **0.91±0.80** | **1.59±0.51** | **0.40±0.31** | **1.16±0.66** | **0.56±0.28** | **1.62±0.65** | **1.93±1.10** | **11.80±13.91** | **1.69±2.38** | **0.90±0.26** | **0.53±0.09** |



**Table 5** Mass concentrations of PE PM$_{2.5}$-bound PAHs, PAEs and hopanes species for women at DF, students at WB and drivers at MT (ng m$^{-3}$).

| Specific species (abbreviation) | Women at DF | | Students at WB | | Drivers at MT | |
|---|---|---|---|---|---|---|
| | Average | Stdev* | Average | Stdev* | Average | Stdev* |
| acenaphthene (ACE) | 0.4 | 0.5 | 0.6 | 1.2 | 0.7 | 1.7 |
| fluorene (FLO) | 0.3 | 0.3 | 0.3 | 0.6 | 0.1 | 0.0 |
| phenanthrene (PHE) | 0.8 | 0.4 | 0.9 | 1.2 | 0.6 | 0.1 |
| anthracene (ANT) | 0.3 | 0.2 | 0.2 | 0.2 | 0.2 | 0.0 |
| fluoranthene (FLU) | 1.0 | 0.4 | 1.0 | 0.7 | 0.6 | 0.1 |
| pyrene (PYR) | 1.2 | 0.5 | 1.0 | 0.5 | 0.6 | 0.1 |
| benzo[a]anthracene (BaA) | 4.5 | 8.5 | 2.2 | 1.5 | 1.1 | 0.5 |
| chrysene (CHR) | 6.1 | 11.2 | 3.0 | 1.6 | 1.8 | 0.8 |
| benzo[b]fluoranthene (BbF) | 11.6 | 19.2 | 5.6 | 2.7 | 3.6 | 1.2 |
| benzo[k]fluoranthene (BkF) | 4.9 | 4.2 | 5.0 | 2.9 | 3.3 | 1.1 |
| benzo[a]fluoranthene (BaF) | 3.8 | 5.3 | 2.1 | 2.4 | 1.5 | 0.8 |
| benzo[e]pyrene (BeP) | 7.7 | 8.1 | 5.0 | 2.5 | 3.6 | 0.7 |
| benzo[a]pyrene (BaP) | 9.7 | 12.5 | 5.5 | 5.7 | 3.5 | 1.6 |
| perylene (PER) | 2.8 | 5.0 | 1.3 | 1.4 | 0.8 | 0.4 |
| indeno[1,2,3-cd]pyrene (IcdP) | 9.4 | 9.3 | 6.4 | 4.5 | 4.5 | 0.7 |
| benzo[ghi]perylene (BghiP) | 7.8 | 6.1 | 6.0 | 3.6 | 6.4 | 0.5 |
| dibenzo[a,h]anthracene (DahA) | 1.8 | 2.2 | 1.0 | 0.6 | 0.6 | 0.1 |
| coronene (COR) | 2.8 | 1.6 | 2.3 | 1.4 | 3.3 | 0.4 |
| dibenzo[a,e]pyrene (DaeP) | 0.7 | 0.7 | 0.5 | 0.3 | 0.3 | 0.1 |
| **ΣPAHs** | **77.4** | **47.9** | **49.9** | **30.7** | **37.0** | **7.4** |
| dimethyl phthalate (DMP) | 2.2 | 1.0 | 9.6 | 27.9 | 1.9 | 0.5 |
| diethyl phthalate (DEP) | 8.3 | 4.1 | 146.5 | 517.0 | 6.8 | 1.4 |
| di-n-butyl phthalate (DBP) | 224.8 | 90.6 | 440.7 | 848.4 | 248.2 | 42.1 |
| benzyl butyl phthalate (BBP) | 13.8 | 4.3 | 19.7 | 37.3 | 8.1 | 2.9 |
| bis(2-ethylhexyl)phthalate (DEHP) | 566.4 | 181.4 | 688.0 | 899.1 | 376.3 | 144.5 |
| di-n-octyl phthalate (DNOP) | 40.9 | 16.9 | 43.8 | 26.2 | 33.0 | 31.0 |
| bis(2-ethylhexyl)adipate (DEHA) | 25.6 | 6.0 | 32.0 | 41.8 | 23.8 | 19.0 |
| **ΣPAEs** | **882.0** | **193.3** | **1380.4** | **335.2** | **698.1** | **192.4** |
| 17α(H)-22,29,30-trisnorhopane (Tm) | 1.3 | 0.5 | 1.3 | 1.9 | 2.5 | 0.5 |
| 17α(H)-21β(H),30-norhopane (αβ-NH) | 4.0 | 1.2 | 3.3 | 4.1 | 10.6 | 1.9 |
| 17β(H)-21α(H),30-norhopane (βα-NH) | 1.5 | 1.8 | 1.1 | 1.5 | 1.9 | 0.3 |
| 17α(H)-21β(H)-hopane (αβ- HH) | 4.3 | 1.9 | 3.6 | 5.4 | 11.5 | 2.2 |
| 17α(H)-21α(H)-hopane (αα-HH) | 0.8 | 0.2 | 1.0 | 2.0 | 3.6 | 2.1 |
| 17β(H)-21α(H)-hopane (βα-HH) | 0.7 | 0.2 | 0.8 | 1.2 | 2.9 | 1.2 |
| 17α(H)-21β(H),(22S)-homohopane (αβ-S-HH) | 2.3 | 0.7 | 2.2 | 2.4 | 8.9 | 1.3 |
| 17α(H)-21β(H),(22R)-homohopane (αβ-R-HH) | 2.2 | 0.8 | 2.1 | 2.1 | 8.9 | 1.3 |
| **Σhopanes** | **17.1** | **6.4** | **15.6** | **6.1** | **50.9** | **7.9** |

*: standard deviation



**Table 6** Non-cancer risks of heavy metals and cancer risks of PAHs and PAEs via inhalation exposure way in PE $PM_{2.5}$ of women at DF, students at WB and drivers at MT in dry and wet seasons.

| | Dry season | | | Wet season | | |
|---|---|---|---|---|---|---|
| | **Women** | **Students** | **Drivers** | **Women** | **Students** | **Drivers** |
| **Non-cancer risk** | | | | | | |
| Mn | 5.71E-03 | 2.02E-02 | 1.09E-02 | 4.83E-03 | 2.31E-02 | 4.26E-03 |
| Ni | 1.44E-04 | 5.60E-04 | 1.77E-04 | 4.49E-04 | 2.59E-03 | 2.00E-04 |
| Zn | 1.45E-04 | 2.15E-04 | 6.16E-05 | 1.24E-04 | 5.45E-04 | 5.05E-05 |
| Pb | 1.75E-03 | 5.98E-03 | 9.33E-04 | 2.97E-03 | 2.95E-02 | 7.75E-04 |
| **HI** | **7.74E-03** | **2.70E-02** | **1.21E-02** | **8.37E-03** | **5.57E-02** | **5.29E-03** |
| **Cancer risk (ILCR)** | | | | | | |
| PAHs ($[BaP]_{eq}$) | 3.13E-06 | 6.43E-06 | 3.22E-06 | 9.33E-06 | 3.68E-06 | 3.42E-06 |
| PAEs (DEHP) | 2.92E-07 | 3.36E-07 | 1.86E-07 | 3.15E-07 | 4.86E-07 | 1.16E-07 |



**Appendix A.**

# 2016

## Assessing Air Pollution Exposures in southern West Africa

## - Questionnaire for Women

**1. Participant name:** _______________

**2. Interviewer name:** _______________

**3. Sampling site:** _________________

**4. Address of the interview place:** ___________________________________

_________________________________________________________________

**5. Address of the participant home:** __________________________________

_________________________________________________________________

**6. Interview date:** ________/_____/_____ (yyyy/mm/dd)

**7. Interview start time:** _______________     **End time:** _______________

**This questionnaire is for research purposes only. Please think carefully and answer all the questions below. Your answers will be kept completely confidential and your personal information will not be disclosed or displayed in any way and any case.**

**Thank you for your cooperation!**



## A. Basic Information

[Please choose by ⊠ or fill in the answer]

A1. Gender: (0) ☐ Male  (1) ☐ Female

A2. Age: _____ years old

A3. Height: _____ cm; Weight: _____ Kg

A4. Marital status:

(0) ☐ Single  (1) ☐ Married

(2) ☐ Divorced  (3) ☐ Widowed

A5. Highest level of education:

(0) ☐ Primary school

(1) ☐ Junior high school

(2) ☐ High school

(3) ☐ Undergraduate

(4) ☐ Above undergraduate

A6. The total number of family members (including you): _______

A7. Number of adults (18 years or older; including you): _____

A8. Family total annual income: _______ West African Franc / Month

A9. Now professional:

(0) ☐ Unemployed

(1) ☐ Students

(2) ☐ Retired staff

(3) ☐ Workers

(4) ☐ Farmers

(5) ☐ Corporate staff

(6) ☐ Civil servants

(7) ☐ Housewife

(8) ☐ Driver

(9) ☐ Others: _____________

A10. Work address (if any)?

_______________________________

A11. Your housing type:

(0) ☐ Apartment

(1) ☐ One-storey house

(2) ☐ Other: _____________

A12. _________ Floor

A13. Residential area: ______ m²

(0) ☐ One room and one hall

(1) ☐ Two rooms and one hall

(2) ☐ Three rooms and two halls

(3) ☐ Others

A14. How long did you move in this house after it was decorated?

(0) ☐ < 3 months  (1) ☐ 3-6 months

(2) ☐ 6-12 months  (3) ☐ > 12 months

A15. When was your house built? ______(year)

A16. How many years have you lived in this house? _____ (year)

A17. What material is your house built?

(0) ☐ Brick  (1) ☐ Armored concrete

(2) ☐ Timber  (3) ☐ Other materials

A18. What is the material of the floor in your house?

(0) ☐ Cement  (1) ☐ Marble

(2) ☐ Solid wood  (3) ☐ Composite wood

(4) ☐ Tile  (5) ☐ Plastic  (6) ☐ Rock

(7) ☐ Brick  (8) ☐ Bare soil

A19. What is the material of the furniture in your house?

(0) ☐ Solid wood  (1) ☐ Plastic

(2) ☐ Leather  (3) ☐ Metal

(4) ☐ Stone  (5) ☐ Glass

(6) ☐ Cloth  (7) ☐ Artificial board

A20. Has your house been decorated in the last year?

(0) ☐ Yes  (1) ☐ No

↳A21. What kind of decoration?

(0) ☐ Paint

(1) ☐ Change the floor



**(2)** ☐ **Add new furniture**

**(3)** ☐ **Other: ______________**

**A22. Does your house have ventilation equipment?**

    **(0)** ☐ **Yes**     **(1)** ☐ **No**

    ↳ **A23. What kind of equipment?**

**(Please select all suitable answers)**

**(0)** ☐ **Hanging air conditioner**

**(1)** ☐ **Cabinet air conditioner**

**(2)** ☐ **Ventilator**

**A24. How far is your house from the main road?**

    **(0)** ☐ **<20m**    **(1)** ☐ **20-100m**

    **(2)** ☐ **>100m**

**A25. Do you smoke?**

    **(0)** ☐ **Yes**    **(1)** ☐ **No**

**A26. Do you have a smoking history?**

    **(0)** ☐ **Yes**    **(1)** ☐ **No**

    ↳**A27. How long is your smoking history? ______ years**

**A28. Does your family member smoke (not including you)?**

    **(0)** ☐ **Yes**    **(1)** ☐ **Used to smoke**

    **(2)** ☐ **No**

**A29. In general, are you influenced by second hand smoke in the following environments often?**

    **(0)** ☐ **Your own home**

    **(1)** ☐ **Working environment**

    **(2)** ☐ **Other' house**

    **(3)** ☐ **Restaurants, bars, supermarkets, streets and so on**

    **(4)** ☐ **Other: __________**

    **(5)** ☐ **Rarely affected by second hand smoke**

**A30. Do you often drink alcohol?**

    **(0)** ☐ **Yes**  **(1)** ☐ **No**  **(2)** ☐ **Not often**

**A31. Drinking type:**

    **(0)** ☐ **Alcohol**    **(1)** ☐ **Beer**

    **(2)** ☐ **Wine**    **(3)** ☐ **Other**

**A32. Drinking frequency per week:**

    **(0)** ☐ **< once**  **(1)** ☐ **1-3 times**

    **(2)** ☐ **> 3 times**  **(3)** ☐ **I don't know**

**A33. Please describe your health status in general.**

    **(0)** ☐ **Very good**    **(1)** ☐ **Good**

    **(2)** ☐ **Not bad**    **(3)** ☐ **Not good**

**A34. Do you have a family history of allergies?**

**(0)** ☐ **Yes (1)** ☐ **No (2)** ☐ **I don't know**

**A35. Have you been allergic to flowers or animals, food, etc.?**

**(0)** ☐ **Yes (1)** ☐ **No (2)** ☐ **I don't know**

**A36. Have you ever had itchy skin and red patches (rashes) lasting more than 6 months?**

**(0)** ☐ **Yes (1)** ☐ **No (2)** ☐ **I don't know**

**A37. Do your parents have asthma?**

**(0)** ☐ **Yes (1)** ☐ **No (2)** ☐ **I don't know**

**A38. Do you have asthma?**

**(0)** ☐ **Yes (1)** ☐ **No (2)** ☐ **I don't know**

**A39. Have you heard any noise or wheeze in your chest (whistle sound) during breathing?**

**(0)** ☐ **Yes (1)** ☐ **No (2)** ☐ **I don't know**

**A40. Have you had symptoms of sneezing, runny nose or stuffy nose in the absence of a cold?**

**(0)** ☐ **Yes (1)** ☐ **No (2)** ☐ **I don't know**

**A41. Are you diagnosed with high blood pressure by your doctor?**

**(0)** ☐ **Yes (1)** ☐ **No (2)** ☐ **I don't know**

    ↳ **A42. Are you taking antihypertensive drugs every day?**

    **(0)** ☐ **Yes**    **(1)** ☐ **No**

**A43. Are you diagnosed with diabetes by your doctor?**

**(0)** ☐ **Yes (1)** ☐ **No (2)** ☐ **I don't know**



**A44. Are you diagnosed with a myocardial infarction by your doctor?**

**(0)** ☐ **Yes  (1)** ☐ **No (2)** ☐ **I don't know**



## B. Environment

[Please choose by ⊠ or fill in the answer]

**B1. Do you cook at home?**

(0) ☐ Yes   (1) ☐ No

↳**B2. Cooking frequency per day:**

(0) ☐ Once   (1) ☐ Twice

(2) ☐ Three times

(3) ☐ > Three times

**B3. What kind of fuel is used at home for cooking? (Please select all suitable answers)**

(0) ☐ Natural gas

(1) ☐ Coal

(2) ☐ Liquefied petroleum gas (LPG)

(3) ☐ Electricity

(4) ☐ Other: _________

(5) ☐ Don't cook at home

**B4. Does your kitchen have ventilation equipment?**

(0)☐ Yes   (1) ☐ No

↳**B5. What kind of equipment? (Please select all suitable answers)**

(0) ☐ Kitchen smoke exhaust ventilator

(1) ☐ Kitchen ventilator

(2) ☐ Chimney

**B6. Your kitchen area:** _______ m$^2$

**B7. Do you usually make smoked fish?**

(0)☐ Yes   (1) ☐ No

**B8. Where do you usually make smoked fish?**

(0) ☐ Kitchen at home  (1) ☐ Yard at home

(2) ☐ Outdoor-working place

**B9. How many times do you make smoked fish per week?** _______times

**B10. How long do you average make the smoked fish each time?** _______min

**B11. When do you usually make smoked fish?**

(0) ☐ Morning

(1) ☐ Noon

(2) ☐ Afternoon

(3) ☐ Evening

**B12. Do you raise pet at home?**

(0) ☐ Yes   (1) ☐ No

**B13. Do you grow flowers or plants at home?**

(0) ☐ Yes   (1) ☐ No

**B14. Do you use insecticide at home?**

(0) ☐ Yes  (1) ☐ No  (2) ☐ I don't know

**B15. What's the open conditions of your windows at home every day?**

(0) ☐ Wide open < 1h

(1) ☐ Wide open > 3h

(2) ☐ Half open < 1h

(3) ☐ Half open > 1h

(4) ☐ Never open

**B16. What tool do you use to clean house?**

(0) ☐ Broom and mop

(1) ☐ Electric dust collector

## C. Travel habits

[Please choose by ⊠ or fill in the answer]





**C1. How much time do you spend indoors per day, except sleeping?**

(0) ☐ > 50%     (1) ☐ = 50%     (2) ☐ < 50%

**C2. How much sleep do you have daily, including daytime and nighttime? _______ h**

**C3. What time of the day do you stay in your house in general? (Please select all suitable answers)**

| Morning | | Afternoon | | | Evening | | | | |
|---|---|---|---|---|---|---|---|---|---|
| 8-10 am | 10-12 am | 12-14 pm | 14-16 pm | 16-18 pm | 18-20 pm | 20-22 pm | 22-24 pm | Next day 0-4 am | Nexr day 4-8 am |
| | | | | | | | | | |

**C4. What time of the day do you stay at your working place in general? (Please select all suitable answers)**

| Morning | | Afternoon | | | Evening | | | | |
|---|---|---|---|---|---|---|---|---|---|
| 8-10 am | 10-12 am | 12-14 pm | 14-16 pm | 16-18 pm | 18-20 pm | 20-22 pm | 22-24 pm | Next day 0-4 am | Nexr day 4-8 am |
| | | | | | | | | | |

**C5. What is your main travel style when you travel < 3 km from your house?**

(0) ☐ Walk          (1) ☐ Bicycle          (2) ☐ Motorcycle

(3) ☐ Public bus      (4) ☐ Car          (5) ☐ Seldom travel

**C6. How many hours do you spend on traveling each day?**

(0) ☐ 0 h   (1) ☐ 0-0.5 h   (2) ☐ 0.5-1 h   (3) ☐ 1-1.5 h

(4) ☐ 1.5-2 h   (5) ☐ 2-3 h   (6) ☐ > 3 h

**C7. How often do you perform outdoor exercises for longer than 30 minutes per week? _________times**

**C8. What kind of transportation do you use when you go to work (if any)? How long does it take?**

(0) ☐ Walk                    average time _______ min

(1) ☐ Bicycle, tricycle              average time _______ min

(2) ☐ Electric bicycles, motorcycles        average time _______ min

(3) ☐ Bus, private car, taxi            average time _______ min

**The interview is over. Thank you for your cooperation again!**





**Appendix B.**

## 2016

## Assessing Air Pollution Exposures in southern West Africa

## - Questionnaire for Students

**1. Participant name: _______________**

**2. Interviewer name: _______________**

**3. Sampling site: _________________**

**4. Address of the interview place: ___________________________________**

**___________________________________________________________________**

**5. Address of the participant home: _______________________________**

**___________________________________________________________________**

**6. Interview date: ________/_____/_____ (yyyy/mm/dd)**

**7. Interview start time: _______________       End time: _______________**

**This questionnaire is for research purposes only. Please think carefully and answer all the questions below. Your answers will be kept completely confidential and your personal information will not be disclosed or displayed in any way and any case.**

**Thank you for your cooperation!**





## A. Basic Information

[Please choose by ⊠ or fill in the answer]

**A1. Gender: (0)** ☐ **Male (1)** ☐ **Female**

**A2. Age:** _____ **years old**

**A3. Height:** _____ **cm**

**A4. Weight:** _____ **Kg**

**A5. Grade:** _____**grade**

**A6. The total number of family members (including you):** _______

**A7. Number of adults (18 years or older):** _____

**A8. Your housing type:**

 **(3)** ☐ **Apartment**

 **(4)** ☐ **One-storey house**

 **(5)** ☐ **Other:** _____________

 ↳ **A9.** _______ **Floor**

**A10. Residential area:** _____ **m²**

 **(4)** ☐ **One room and one hall**

 **(5)** ☐ **Two rooms and one hall**

 **(6)** ☐ **Three rooms and two halls**

 **(7)** ☐ **Others**

**A11. How many years have you lived in this house?** _____ **(year)**

**A12. What material is your house built?**

 **(0)** ☐ **Brick (1)** ☐ **Armored concrete**

 **(2)** ☐ **Timber (3)** ☐ **Other materials**

**A13. What is the material of the floor in your house?**

 **(0)** ☐ **Cement (1)** ☐ **Marble**

 **(2)** ☐ **Solid wood (3)** ☐ **Composite wood**

 **(4)** ☐ **Tile (5)** ☐ **Plastic (6)** ☐ **Rock**

 **(7)** ☐ **Brick (8)** ☐ **Bare soil**

**A14. What is the material of the furniture in your house?**

 **(0)** ☐ **Solid wood (1)** ☐ **Plastic**

 **(2)** ☐ **Leather (3)** ☐ **Metal**

 **(4)** ☐ **Stone (5)** ☐ **Glass**

 **(6)** ☐ **Cloth (7)** ☐ **Artificial board**

**A15. Has your house been decorated in the last year?**

 **(1)** ☐ **Yes (1)** ☐ **No**

 ↳**A16. What kind of decoration?**

 **(0)** ☐ **Paint**

 **(1)** ☐ **Change the floor**

 **(2)** ☐ **Add new furniture**

 **(3)** ☐ **Other:** ___________

**A17. Does your house have ventilation equipment?**

 **(1)** ☐ **Yes (1)** ☐ **No**

 ↳**A18. What kind of equipment? (Please select all suitable answers)**

 **(0)** ☐ **Hanging air conditioner**

 **(1)** ☐ **Cabinet air conditioner**

 **(2)** ☐ **Ventilator**

**A19. How far is your house from the main road?**

 **(0)** ☐ **<20m (1)** ☐ **20-100m**

 **(2)** ☐ **>100m**

**A20. Does your classroom have ventilation equipment?**

 **(0)** ☐ **Yes (1)** ☐ **No**

 ↳**A21. What kind of equipment? (Please**



select all suitable answers)

(0) ☐ Hanging air conditioner

(1) ☐ Cabinet air conditioner

(2) ☐ Ventilator

**A22. How far is your classroom from the main road?**

(0) ☐ <20m    (1) ☐ 20-100m

(2) ☐ >100m

**A23. Do you smoke?**

(0) ☐ Yes    (1) ☐ No

**A24. Does your family member smoke (not including you)?**

(0) ☐ Yes    (1) ☐ Used to smoke

(2) ☐ No

**A25. In general, are you influenced by second hand smoke in the following environments often?**

(0) ☐ Your own home

(1) ☐ Working environment

(2) ☐ Other' house

(3) ☐ Restaurants, bars, supermarkets, streets and so on

(4) ☐ Other: __________

(5) ☐ Rarely affected by second hand smoke

**A26. Do you often drink alcohol?**

(0) ☐ Yes  (1) ☐ No  (2) ☐ Not often

**A27. Mainly drinking type:**

(0) ☐ Alcohol    (1) ☐ Beer

(2) ☐ Wine    (3) ☐ Other

**A28. Please describe your health status in general.**

(0) ☐ Very good    (1) ☐ Good

(2) ☐ Not bad    (3) ☐ Not good

**A29. Do you have a family history of allergies?**

(0) ☐ Yes  (1) ☐ No  (2) ☐ I don't know

**A30. Have you been allergic to flowers or animals, food, etc.?**

(0) ☐ Yes  (1) ☐ No  (2) ☐ I don't know

**A31. Have you ever had itchy skin and red patches (rashes) lasting more than 6 months?**

(0) ☐ Yes  (1) ☐ No  (2) ☐ I don't know

**A32. Do your parents have asthma?**

(0) ☐ Yes  (1) ☐ No  (2) ☐ I don't know

**A33. Do you have asthma?**

(0) ☐ Yes  (1) ☐ No  (2) ☐ I don't know

**A34. Have you heard any noise or wheeze in your chest (whistle sound) during breathing?**

(0) ☐ Yes  (1) ☐ No  (2) ☐ I don't know

**A35. Have you had symptoms of sneezing, runny nose or stuffy nose in the absence of a cold?**

(0) ☐ Yes  (1) ☐ No  (2) ☐ I don't know



## B. Environment

[Please choose by ⊠ or fill in the answer]

**B1. Do you cook at home?**

(0) ☐ Yes    (1) ☐ No

↳**B2. Cooking frequency per day:**

(0) ☐ Once    (1) ☐ Twice

(2) ☐ Three times

(3) ☐ > Three times

**B3. What kind of fuel is used at home for cooking? (Please select all suitable answers)**

(0) ☐ Natural gas

(1) ☐ Coal

(2) ☐ Liquefied petroleum gas (LPG)

(3) ☐ Electricity

(4) ☐ Other: _________

(5) ☐ Don't cook at home

**B4. Does your kitchen have ventilation equipment?**

(0)☐ Yes    (1) ☐ No

↳ **B5. What kind of equipment? (Please select all suitable answers)**

(0) ☐ Kitchen smoke exhaust ventilator

(1) ☐ Kitchen ventilator

(2) ☐ Chimney

**B6. Your kitchen area:** ______ $m^2$

**B7. Does your family usually make smoked fish?**

(0)☐ Yes    (1) ☐ No

**B8. How many times does your family eat smoked fish per week?** ______times.

**B9. Do you raise pet at home?**

(0) ☐ Yes    (1) ☐ No

**B10. Do you grow flowers or plants at home?**

(0) ☐ Yes    (1) ☐ No

**B11. Do you use insecticide at home?**

(0) ☐ Yes (1) ☐ No (2) ☐ I don't know

**B12. What's the open conditions of your windows at home every day?**

(0) ☐ Wide open < 1h

(1) ☐ Wide open > 3h

(2) ☐ Half open < 1h

(3) ☐ Half open > 3h

(4) ☐ Never open

**B13. What's the open conditions of your windows at classroom every day?**

(0) ☐ Wide open < 1h

(1) ☐ Wide open > 3h

(2) ☐ Half open < 1h

(3) ☐ Half open > 3h

(4) ☐ Never open

**B14. What tool does your family use to clean house?**

(2) ☐ Broom and mop

(3) ☐ Electric dust collector

**B15. What tool does your class use to clean classroom?**

(0) ☐ Broom and mop

(1) ☐ Electric dust collector

**B16. How far does dumps away from your house?** ______ m

Walking time:

(0) ☐ < 5min



(1) ☐ 5-10min

(2) ☐ 10-15min

(3) ☐ 15-20min

(4) ☐ 20-30 min

(5) ☐ > 30 min

**B17. How far does dumps away from your school?** _______ m

    **Walking time:**

(0) ☐ < 5min

(1) ☐ 5-10min

(2) ☐ 10-15min

(3) ☐ 15-20min

(4) ☐ 20-30 min

(5) ☐ > 30 min

**B18. Can you see waste burning at home?**

(0) ☐ Yes    (1) ☐ No

    ↳**B19. How many times per week?**

    ______times

**B20. Can you see waste burning at school?**

(0) ☐ Yes    (1) ☐ No

    ↳**B21. How many times per week?**

    ______times

**B22. Can you smell smoke from waste burning at home?**

(0) ☐ Yes    (1) ☐ No

**B23. Can you smell smoke from waste burning at school?**

## C. Travel habits

[Please choose by ☒ or fill in the answer]

**C1. How much time do you spend indoors per day, except sleeping?**

(0) ☐ Yes    (1) ☐ No

**B24. How to deal with your home waste?**

(0) ☐ Throwing to dumps

(1) ☐ Burning by yourselves

(2) ☐ I don't know

**B25. Does waste burning in dumps impact on your live?**

(0) ☐ Yes  (1) ☐ No

    ↳**B26. What specific performance?**

(Please select all suitable answers)

(0) ☐  Road congestion, inconvenient travel

(1) ☐ Air odor, black smoke filled

(2) ☐ Air pollution, low visibility

(3) ☐  Water pollution, fish and shrimp death

**B27. Does waste burning in dumps impact on your health?**

(0) ☐ Yes  (1) ☐ No

    ↳**B28. What specific performance?**

(Please select all suitable answers)

(0) ☐ Congestion, runny nose

(1) ☐ Dry eyes, tears

(2) ☐ Skin allergies

(3) ☐ Throat dry, inflamed

(4) ☐ Difficulty breathing

(5) ☐ Other: ________________



(0) ☐ > 50%     (1) ☐ = 50%     (2) ☐ < 50%

**C2. How much sleep do you have daily, including daytime and nighttime? _______ h**

**C3. What time of the day do you stay in your house in general? (Please select all suitable answers)**

| Morning | | Afternoon | | | Evening | | | | |
|---|---|---|---|---|---|---|---|---|---|
| 8-10 am | 10-12 am | 12-14 pm | 14-16 pm | 16-18 pm | 18-20 pm | 20-22 pm | 22-24 pm | Next day 0-4 am | Nexr day 4-8 am |
| | | | | | | | | | |

**C4. What time of the day do you stay at school in general? (Please select all suitable answers)**

| Morning | | Afternoon | | | Evening | | | | |
|---|---|---|---|---|---|---|---|---|---|
| 8-10 am | 10-12 am | 12-14 pm | 14-16 pm | 16-18 pm | 18-20 pm | 20-22 pm | 22-24 pm | Next day 0-4 am | Nexr day 4-8 am |
| | | | | | | | | | |

**C5. What is your main travel style when you travel < 3 km from your house?**

(0) ☐ **Walk**          (1) ☐ **Bicycle**          (2) ☐ **Motorcycle**

(3) ☐ **Public bus**          (4) ☐ **Car**          (5) ☐ **Seldom travel**

**C6. How many hours do you spend on traveling each day?**

(0) ☐ **0 h**   (1) ☐ **0-0.5 h**   (2) ☐ **0.5-1 h**   (3) ☐ **1-1.5 h**

(4) ☐ **1.5-2 h**   (5) ☐ **2-3 h**   (6) ☐ **> 3 h**

**C7. How often do you perform outdoor exercises for longer than 30 minutes per week? ________ times**

**C8. What kind of transportation do you use when you go to school? How long does it take?**

(0) ☐ **Walk**                    average time _______ min

(1) ☐ **Bicycle, tricycle**                average time _______ min

(2) ☐ **Electric bicycles, motorcycles**        average time _______ min

(3) ☐ **Bus, private car, taxi**            average time _______ min

**The interview is over. Thank you for your cooperation again!**



**Appendix C.**

# 2016

## Assessing Air Pollution Exposures in southern West Africa

## - Questionnaire for Drivers

**1. Participant name: _______________**

**2. Interviewer name: _______________**

**3. Sampling site: __________________**

**4. Address of the interview place: ______________________________**

**______________________________________________________________**

**5. Address of the participant home: ____________________________**

**______________________________________________________________**

**6. Interview date: ________/_____/_____ (yyyy/mm/dd)**

**7. Interview start time: _______________   End time: _______________**

**This questionnaire is for research purposes only. Please think carefully and answer all the questions below. Your answers will be kept completely confidential and your personal information will not be disclosed or displayed in any way and any case.**

**Thank you for your cooperation!**




## A. Basic Information

[Please choose by ⊠ or fill in the answer]

**A1.** Gender: (0) ☐ Male  (1) ☐ Female

**A2.** Age: _____ years old

**A3.** Height: _____ cm; Weight: _____ Kg

**A4.** Marital status:

(0) ☐ Single    (1) ☐ Married

(2) ☐ Divorced    (3) ☐ Widowed

**A5.** Highest level of education:

(5) ☐ Primary school

(6) ☐ Junior high school

(7) ☐ High school

(8) ☐ Undergraduate

(9) ☐ Above undergraduate

**A6.** The total number of family members (including you): _______

**A7.** Number of adults (18 years or older; including you): _____

**A8.** Family total annual income: _______ West African Franc / Month

**A9.** Which kind of car do you drive when you work?

(0) ☐ Motorcycle  (1) ☐ Car

(2) ☐ Tricycle    (3) ☐ Others_________

**A10.** As a driver, how long do you work per day?

(0) ☐ < 1h

(1) ☐ 1-3h

(2) ☐ 4-6h

(3) ☐ 7-9h

(4) ☐ 10-12h

(5) ☐ >12h

**A11.** Your housing type:

(6) ☐ Apartment

(7) ☐ One-storey house

(8) ☐ Other: _____________

↳**A12.** ________ Floor

**A13.** Residential area: ______ m²

(8) ☐ One room and one hall

(9) ☐ Two rooms and one hall

(10) ☐ Three rooms and two halls

(11) ☐ Others

**A14.** How long did you move in this house after it was decorated?

(0) ☐ < 3 months    (1) ☐ 3-6 months

(2) ☐ 6-12 months    (3) ☐ > 12 months

**A15.** When was your house built? ______(year)

**A16.** How many years have you lived in this house? _____ (year)

**A17.** What material is your house built?

(0) ☐ Brick    (1) ☐ Armored concrete

(2) ☐ Timber    (3) ☐ Other materials

**A18.** What is the material of the floor in your house?

(0) ☐ Cement    (1) ☐ Marble

(2) ☐ Solid wood  (3) ☐ Composite wood

(4) ☐ Tile   (5) ☐ Plastic  (6) ☐ Rock

(7) ☐ Brick    (8) ☐ Bare soil

**A19.** What is the material of the furniture in your house?

(0) ☐ Solid wood    (1) ☐ Plastic

(2) ☐ Leather    (3) ☐ Metal

(4) ☐ Stone    (5) ☐ Glass

(6) ☐ Cloth    (7) ☐ Artificial board

**A20.** Has your house been decorated in the last year?

(2) ☐ Yes    (1) ☐ No

↳**A21.** What kind of decoration?

(0) ☐ Paint

(1) ☐ Change the floor

(2) ☐ Add new furniture



(3) ☐ Other: _____________

**A22. Does your house have ventilation equipment?**

 (2) ☐ Yes    (1) ☐ No

  ↳ **A23. What kind of equipment?**

**(Please select all suitable answers)**

(0) ☐ Hanging air conditioner

(1) ☐ Cabinet air conditioner

(2) ☐ Ventilator

**A24. How far is your house from the main road?**

 (0) ☐ < 20m    (1) ☐ 20-100m

 (2) ☐ > 100m

**A25. Do you smoke?**

 (0) ☐ Yes    (1) ☐ No

**A26. Do you have a smoking history?**

 (0) ☐ Yes    (1) ☐ No

  ↳**A27. How long is your smoking history?** _____ (year)

**A28. Does your family member smoke (not including you)?**

 (0) ☐ Yes    (1) ☐ Used to smoke

 (2) ☐ No

**A29. In general, are you influenced by second hand smoke in the following environments often?**

 (0) ☐ Your own home

 (1) ☐ Working environment

 (2) ☐ Other' house

 (3) ☐ Restaurants, bars, supermarkets, streets and so on

 (4) ☐ Other: __________

 (5) ☐ Rarely affected by second hand smoke

**A30. Do you often drink alcohol?**

 (0) ☐ Yes    (1) ☐ No

 (2) ☐ Not often  (3) ☐Already abstaining

**A31. Mainly drinking type:**

 (0) ☐ Alcohol  (1) ☐ Beer

 (2) ☐ Wine   (3) ☐ Other

**A32. Drinking frequency per week (If any):** _______times.

**A33. Please describe your health status in general.**

 (0) ☐ Very good  (1) ☐ Good

 (2) ☐ Not bad   (3) ☐ Not good

**A34. Do you have a family history of allergies?**

(0) ☐ Yes (1) ☐ No (2) ☐ I don't know

**A35. Have you been allergic to flowers or animals, food, etc.?**

(0) ☐ Yes (1) ☐ No (2) ☐ I don't know

**A36. Have you ever had itchy skin and red patches (rashes) lasting more than 6 months?**

(0) ☐ Yes (1) ☐ No (2) ☐ I don't know

**A37. Do your parents have asthma?**

(0) ☐ Yes (1) ☐ No (2) ☐ I don't know

**A38. Do you have asthma?**

(0) ☐ Yes (1) ☐ No (2) ☐ I don't know

**A39. Have you heard any noise or wheeze in your chest (whistle sound) during breathing?**

(0) ☐ Yes (1) ☐ No (2) ☐ I don't know

**A40. Have you had symptoms of sneezing, runny nose or stuffy nose in the absence of a cold?**

(0) ☐ Yes (1) ☐ No (2) ☐ I don't know

**A41. Are you diagnosed with high blood pressure by your doctor?**

(0) ☐ Yes (1) ☐ No (2) ☐ I don't know

 ↳ **A42. Are you taking antihypertensive drugs every day?**

  (0) ☐ Yes    (1) ☐ No

**A43. Are you diagnosed with diabetes by your doctor?**

(0) ☐ Yes (1) ☐ No (2) ☐ I don't know

**A44. Are you diagnosed with a myocardial**



**infarction by your doctor?**

**(0)** ☐ **Yes  (1)** ☐ **No (2)** ☐ **I don't know**



## B. Environment

[Please choose by ⊠ or fill in the answer]

**B1. Do you cook at home?**

(0) ☐ Yes  (1) ☐ No

↳**B2. Cooking frequency per day:**

(0) ☐ Once   (1) ☐ Twice

(2) ☐ Three times

(3) ☐ > Three times

**B3. What kind of fuel is used at home for cooking? (Please select all suitable answers)**

(0) ☐ Natural gas

(1) ☐ Coal

(2) ☐ Liquefied petroleum gas (LPG)

(3) ☐ Electricity

(4) ☐ Other: _______

(5) ☐ Don't cook at home

**B4. Does your kitchen has ventilation equipment?**

(0) ☐ Yes     (1) ☐ No

↳ **B5. What kind of equipment? (Please select all suitable answers)**

(0) ☐ Kitchen smoke exhaust ventilator

(1) ☐ Kitchen ventilator

(2) ☐ Chimney

**B6. Your kitchen area: _______ m$^2$**

**B7. Do you cook at home?**

(0) ☐ Yes     (1) ☐ No

**B8. Does your family usually make smoked fish?**

(0) ☐ Yes     (1) ☐ No

**B9. How many times do you eat smoked fish at home per week? _______times.**

**B10. Do you raise pet at home?**

(0) ☐ Yes       (1) ☐ No

**B11. Do you grow flowers or plants at home?**

(0) ☐ Yes      (1) ☐ No

**B12. Do you use insecticide at home?**

(0) ☐ Yes  (1) ☐ No  (2) ☐ I don't know

**B13. What's the open conditions of your windows at home every day?**

(0) ☐ Wide open < 1h

(1) ☐ Wide open > 3h

(2) ☐ Half open < 1h

(3) ☐ Half open > 1h

(4) ☐ Never open

**B14. What tool do you use to clean house?**

(4) ☐ Broom and mop

(5) ☐ Electric dust collector

**B15. What kind of road do you usually drive on? (Please select all suitable answers)**

(0) ☐ Unsurfaced road (1) ☐ Stone road

(2) ☐ Asphalt road  (3) ☐ Cement road

(4) ☐ Others: _______

**B16. What kind of environment do you usually drive on? (Please select all suitable answers)**

(0) ☐ Business district (1) ☐ Industrial area

(2) ☐ Residential area  (3) ☐ Suburbs

(4) ☐ Others: _______

**B17. What type of power does your motorcycle use?**

(0) ☐ Diesel     (1) ☐ Gasoline

(2) ☐ Manpower  (3) ☐ Electricity

**B18. When do you usually work with motorcycle?**





(0) ☐ Daytime  (1) ☐ Nighttime

(2) ☐ Morning rush hour

(3) ☐ Night rush hour

**B19. What is the main purpose of your driving motorcycle?**

(0) ☐ Freight  (1) ☐ Passenger

(2) ☐ Both of above (3) ☐ Others______

**B20. How far do you drive motorcycle per day?**

______Km

**B21. How do you feel about the surrounding air**

when driving every day?

(0) ☐ Very good  (1) ☐ Good

(2) ☐ Not bad  (3) ☐ Bad

**B22. Can you smell the exhaust when driving every day?**

(0)☐ Yes  (1) ☐ No

**B23. Do you wear a helmet when driving every day (if applicable)?**

(0)☐ Yes  (1) ☐ No

## C. Travel habits

[Please choose by ☒ or fill in the answer]

**C1. How much time do you spend indoors per day, except sleeping?**

(0) ☐ > 50%  (1) ☐ = 50%  (2) ☐ < 50%

**C2. How much sleep do you have daily, including daytime and nighttime? ______ h**

**C3. What time of the day do you stay in your house in general? (Please select all suitable answers)**

| Morning | | Afternoon | | | Evening | | | | |
|---|---|---|---|---|---|---|---|---|---|
| 8-10 am | 10-12 am | 12-14 pm | 14-16 pm | 16-18 pm | 18-20 pm | 20-22 pm | 22-24 pm | Next day 0-4 am | Nexr day 4-8 am |
| | | | | | | | | | |

**C4. What time of the day do you drive cars for work in general? (Please select all suitable answers)**

| Morning | | Afternoon | | | Evening | | | | |
|---|---|---|---|---|---|---|---|---|---|
| 8-10 am | 10-12 am | 12-14 pm | 14-16 pm | 16-18 pm | 18-20 pm | 20-22 pm | 22-24 pm | Next day 0-4 am | Nexr day 4-8 am |
| | | | | | | | | | |

**C5. What is your main travel style when you travel < 3 km from your house?**

(0) ☐ Walk  (1) ☐ Bicycle  (2) ☐ Motorcycle

(3) ☐ Public bus (4) ☐ Car  (5) ☐ Seldom travel

**C6. How often do you perform outdoor exercises for longer than 30 minutes per week?**

________times

**The interview is over. Thank you for your cooperation again!**





**Appendix D.**

2016

Assessing Air Pollution Exposures in
southern West Africa

## DETAILED TIME-ACTIVITY DIARY

Participant name: _ _ _ _ _ _ _ _

Start date: _ _ / _ _ / _ _     Time: _ _:_ _
         MM/DD/YY

End date: _ _ / _ _ / _ _     Time: _ _:_ _
         MM/DD/YY

Interviewer name: _ _ _ _ _ _ _ _



**Atmospheric Chemistry and Physics** Open Access
Discussions

## DETAILED ACTIVITY DIARY FOR WOMAN

| Time | Location (mark all that apply) | Activities (mark all that apply) |
|---|---|---|
| x:00 am-x:30 am*<br><br>Pollution sources<br>☐ Environmental tobacco smoke<br>☐ Cooking<br>  ☐ smoking fish<br>  ☐ no smoking fish<br>☐ Use of cleaning products<br>☐ House decoration<br>☐ Transportation emissions<br>☐ Other (specify):<br>________________ | Indoors<br>☐ Home<br>  ☐ kitchen<br>  ☐ living room<br>  ☐ bedroom<br>  ☐ courtyard<br>☐ Work<br>☐ Other (specify):<br>________________<br><br>Outdoors<br>☐ In transit (specify):<br>________________<br>☐ Other (specify):<br>________________ | ☐ Cooking<br>☐ Smoking fish<br>☐ Cleaning room<br>☐ Washing the clothes<br>☐ Taking food<br>☐ Watching television<br>☐ Taking a rest<br>☐ Working at office<br>☐ Going out<br>☐ Taking exercise<br>☐ Shopping<br>☐ Visiting friends<br>☐ Other (specify):<br>________________ |

*X refers to the hour.



## DETAILED ACTIVITY DIARY FOR STUDENT

| Time | Location (mark all that apply) | Activities (mark all that apply) |
|---|---|---|
| x:00 am-x:30 am*<br><br>Pollution sources<br>☐ Environmental tobacco smoke<br>☐ Cooking<br>  ☐ smoking fish<br>  ☐ no smoking fish<br>☐ Use of cleaning products<br>☐ House decoration<br>☐ Transportation emissions<br>☐ Other (specify):<br>_____________ | Indoors<br>☐ Home<br>  ☐ kitchen<br>  ☐ living room<br>  ☐ bedroom<br>  ☐ courtyard<br>☐ School classroom<br>☐ Other (specify):<br>_____________<br><br>Outdoors<br>☐ In transit (specify):<br>_____________<br>☐ Other (specify):<br>_____________ | ☐ Cooking<br>☐ Smoking fish<br>☐ Cleaning room<br>☐ Washing the clothes<br>☐ Taking food<br>☐ Watching television<br>☐ Taking a rest<br>☐ Studying at school<br>☐ Studying at home<br>☐ Going out<br>☐ Taking exercise<br>☐ Shopping<br>☐ Playing<br>☐ Other (specify):<br>_____________ |

*X refers to the hour.



## DETAILED ACTIVITY DIARY FOR DRIVER

| Time | Location (mark all that apply) | Activities (mark all that apply) |
|---|---|---|
| x:00 am-x:30 am*<br><br>Pollution sources<br>☐ Environmental tobacco smoke<br>☐ Cooking<br> ☐ smoking fish<br> ☐ no smoking fish<br>☐ Use of cleaning products<br>☐ House decoration<br>☐ Transportation emissions<br>☐ Other (specify):<br>__________ | Indoors<br>☐ Home<br> ☐ kitchen<br> ☐ living room<br> ☐ bedroom<br> ☐ courtyard<br>☐ Other (specify):<br>__________<br><br>Outdoors<br>☐ In transit (specify):<br>__________<br>☐ In the street<br>☐ Other (specify):<br>__________ | ☐ Cooking<br>☐ Smoking fish<br>☐ Cleaning room<br>☐ Washing the clothes<br>☐ Taking food<br>☐ Watching television<br>☐ Taking a rest<br>☐ Driving the MOTO<br>☐ Driving the car<br>☐ Going out<br>☐ Taking exercise<br>☐ Shopping<br>☐ Visiting friends<br>☐ Other (specify):<br>__________ |

*X refers to the hour.

