# Peer review of "Personal exposure to PM2.5 emitted from typical anthropogenic sources in"

_Atmospheric Chemistry and Physics, 2018_

## Referee Comment (RC1) · Anonymous Referee #1 · 11 Nov 2018

Review of 'Personal exposure to PM2.5 emitted from typical anthropogenic sources in Southern West Africa (SWA): Chemical characteristics and associated health risks' by H. Xu et al.

Megacities are developing rapidly in the developing world and South West Africa is no exception. This rapid urbanization is putting huge demands on the ill-adapted infrastructure. A major matter of concern for the local populations and policy-makers is the quantification of the impact of poor air-quality on human health. When they exist, the measurements performed by the stations of the air-quality monitoring networks provide

a good estimate of the ambient pollutants concentrations. However, because of their individual activities a significant fraction of the population is thought to be exposed to concentrations much larger than suggested by these routine measurements. In South West Africa, data are cruelly lacking and this question is still largely open. Therefore, the authors pursue in their present work three different objectives. First, they want to quantify the personal exposure to PM2.5 of women working in an outdoor grilling place, students living close to a landfill in which waste is being burnt, and taxi drivers who spend long working hours in the middle of a very dense traffic. The participants to the study were equipped with personal sampling devices that they carried continuously for two different weeks (one in the dry, and one the wet, seasons). The daily samples collected during these periods were carefully analyzed in the lab, and their composition determined. Finally, an attempt is made to assess the health risk resulting from this exposure to large pollutants concentrations. My opinion is that with its very important and quite novel subject this work has the potential to become a very interesting one. However, it currently suffers from several flaws (see below) that need being addressed before it can be accepted for publication.

Comments and suggestions for improvement: - I understand that the authors are not native speakers but in some parts of the manuscript, the clumsy phrasing hinders comprehension. This point should be taken care of. - In the chemical analysis, I am surprised by the choice of Fe as a tracer of the crustal component of the aerosol. It is well known that at least a part of its concentration is contributed by anthropogenic activities. Wouldn't Al or Ca be a better choice? By the way, why were these elements not quantified by the XRF analysis? - In the health risk assessment, it would be useful to detail the type of risk quantified. The categories 'cancer-risk' and 'non cancer-risk' are very broad. Also, is the risk a long-term or a short-term one? Why did you assess only the risks resulting from exposure to Mn, Ni, Zn, Pb, the PAHs and the PAEs? There is also a risk due to exposure to PM2.5 and given the large concentrations reported in your work, I expect this one might be very important. - In the results section (line 328-330), you cannot extrapolate to the whole SWA region your results collected during two

weeks at three very specific locations. - Line 400: you say that total carbon was the most important chemical species in PE PM2.5 but it contributes only about 20% to the mass concentration. Isn't this contradictory? What about mineral dust? - Paragraph 705-724: First, you say that there is no non-carcinogenic risk linked with the exposure to Mn, Pb, Ni, and Zn (line 709), then you discuss the fact that the risk is much higher in the dry season (line 718). What is the point of discussing the magnitude of this risk, especially before repeating (line 723-724) that it is negligible?
* * *

---

## Author Comment (AC1) · 16 Nov 2018

We would like to thank the reviewer for all the comments firstly. Below we address to the best of our ability each comment.

Comments and suggestions for improvement:

- I understand that the authors are not native speakers but in some parts of the manuscript, the clumsy phrasing hinders comprehension. This point should be taken care of.

Response: I will look for a native speaker cooperator to polish the language of this manuscript.

- In the chemical analysis, I am surprised by the choice of Fe as a tracer of the crustal component of the aerosol. It is well known that at least a part of its concentration is contributed by anthropogenic activities. Wouldn't Al or Ca be a better choice? By the way, why were these elements not quantified by the XRF analysis?

Response: This is a good point. In order to analyze the carbonaceous aerosol, the authors selected quartz fiber filters to collect personal exposure PM2.5 samples in this study. Due to the limitations of personal exposure sampling, it is difficult to collect quartz fiber filter and Teflon membrane filter simultaneously.

Moreover, the analytical uncertainties by using ED-XRF for smaller molecular weight crustal elements in quartz fiber filter (with high Na, Al, Ca and Mg background), such as Al, Si and Ca, are high. So, Al, Si and Ca are not suitable to be used as a tracer of the crustal component of the aerosol in this study. Meanwhile, the high accuracy of Fe analysis with ED-XRF has been demonstrated in our previous publication (Xu at al., 2016), and Fe has been used often as a tracer for crustal component in PM2.5 (e.g., Cao et al., 2004; Hao et al., 2007; Sun et al., 2014; Wu et al., 2012; Xu at al., 2016).

Furthermore, based on the previous references (Gelado-Caballero et al., 2012; Zhuang et al., 2001), the enrichment factors of Fe in dust storm period and non-dust storm were both 1-2, always < 10, proving that Fe in aerosol was still mainly derived from the crustal source. Therefore, taking into account the above points, the author finally picked Fe as a tracer of the crustal component in this study.

Reference:

Cao, J. J., Rong, B., Lee, S. C., Chow, J. C., Ho, K. F., Liu, S. X., and Zhu, C. S.: Composition of indoor aerosols at emperor Qin's terra-cotta museum, Xi'an, China, during summer, China Particuology, 3(3), 170-175, 2004.

Gelado-Caballero, M. D., López-García, P., Prieto, S., Patey, M. D., Collado, C., and Hérnández-Brito, J. J.: Long-term aerosol measurements in Gran Canaria, Canary Islands: Particle concentration, sources and elemental composition, J. Geophy. Res.-Atmos., 117, D03304, doi:10.1029/2011JD016646, 2012.

Hao, Y. C., Guo, Z. G., Yang, Z. S., Fang, M., and Feng, J. L.: Seasonal variations and sources of various elements in the atmospheric aerosols in Qingdao, China, Atmos. Res., 85, 27-37, 2007.

Sun, Y. Y., Hu, X., Wu, J. C., Lian, H. Z., and Chen, Y. J.: Fractionation and health risks of atmospheric particle-bound As and heavy metals in summer and winter, Sci. Total Environ., 493, 487-494, 2014.

Wu, F., Zhang, D. Z., Cao, J. J., Xu, H. M., and An, Z.S: Soil-derived sulfate in atmospheric dust particles at Taklimakan desert, Geophy. Res. Lett., 39, L24803, doi:10.1029/2012GL054406, 2012.

Xu, H.M., Cao, J. J., Chow, J. C., Huang, R.-J., Shen, Z. X., Chen, L. W. A., Ho, K. F., and Watson, J. G..: Inter-annual variability of wintertime PM2.5 chemical composition in Xi'an, China: Evidences of changing source emissions, Sci. Total Environ. 545-546, 546-555, 2016.

Zhuang, G. S., Guo, J. H., Yuan, H., and Zhao, C. Y.: The compositions, sources, and size distribution of the dust storm from China in spring of 2000 and its impact on the global environment, Chinese Sci. Bull., 46(11), 895-900, 2001.

- In the health risk assessment, it would be useful to detail the type of risk quantified. The categories 'cancer risk' and 'non-cancer risk' are very broad. Also, is the risk

a long-term or a short-term one? Why did you assess only the risks resulting from exposure to Mn, Ni, Zn, Pb, the PAHs and the PAEs? There is also a risk due to exposure to PM2.5 and given the large concentrations reported in your work, I expect this one might be very important.

Response: We agree and understand the reviewer's concern. The details of these categories have been clarified and revised in the manuscript: "The heavy metals non-carcinogenic risks and toxic organics carcinogenic risks of PM2.5 via inhalation were calculated according to the U.S. EPA health risk assessment model (USEPA, 2004, 2011).". U.S. EPA health risk assessment model is the process to estimate the nature and probability of adverse health effects in humans who may be exposed to chemicals in contaminated environmental media, now or in the future. The reason we chose Mn, Ni, Zn, Pb, PAHs and PAEs to assess the health risks in personal exposure PM2.5 samples is because these chemicals (among all the chemicals we analyzed in this study) are included in this model and they are assessed to be hazardous to human health in the previous studies (e.g., Hu et al., 2018; Kong et al., 2015; Sun et al., 2014; Xu et al., 2018).

Moreover, indeed, as the reviewer said "There is also a risk due to exposure to PM2.5", but PM2.5 is a complex mixture containing a lot of chemicals. There is no clear and better way to assess its whole health risks for now based on PM2.5 chemical concentrations (except for the model simulation and medical animal exposure experiments). So, at this moment, we calculated the risks of the certain toxic chemicals in PM2.5 to estimate PM2.5 health risks.

Reference:

Hu, Y. J., Bao, L. J., Huang, C. L., Li, S. M., Li, Liu, P., and Zeng, E. Y.: Assessment of airborne polycyclic aromatic hydrocarbons in a megacity of South China: Spatiotemporal variability, indoor-outdoor interplay and potential human health risk, Environ. Pollut., 238, 431-439, 2018.

Kong, S. F., Li, L., Li, X. X., Yin, Y., Chen, K., Liu, D. T., Yuan, L., Zhang, Y. J., Shan, Y. P., and Ji, Y. Q.: The impacts of firework burning at the Chinese Spring Festival on air quality: insights of tracers, source evolution and aging processes, Atmos. Chem. Phys., 15, 2167-2184, 2015.

Sun, Y. Y., Hu, X., Wu, J. C., Lian, H. Z., and Chen, Y. J.: Fractionation and health risks of atmospheric particle-bound As and heavy metals in summer and winter, Sci. Total Environ., 493, 487-494, 2014.

Xu, H. M., Guinot, B., Cao, J. J., Li, Y. Q., Niu, X. Y., Ho, K. F., Shen, Z. X., Liu, S. X., Zhang, T., Lei, Y. L., Zhang, Q., Sun, J., and Gao, J. J.: Source, health risk and composition impact of outdoor very fine particles (VFPs) to school indoor environment in Xi'an, Northwestern China, Sci. Total Environ., 612, 238-246, 2018.

- In the results section (line 328-330), you cannot extrapolate to the whole SWA region your results collected during two weeks at three very specific locations.

Response: Thank you for this point. The authors have revised this statement to "The average personal exposure to PM2.5 (PE PM2.5) mass concentrations were 331.7±190.7, 356.9±71.9 and 242.8±67.6 $\mu$g m-3 for women at Domestic Fires (DF), students at Waste Burning (WB) and drivers at Motorcycle Traffic (MT) respectively in this study." Moreover, the authors have checked related issue in this manuscript and will make changes in the revised version.

- Line 400: you say that total carbon was the most important chemical species in PE PM2.5 but it contributes only about 20

Response: We are sorry for the confusion. In section "3.1.2. PE PM2.5 chemical compositions", the authors were talking about the PE PM2.5 chemical compositions, which includes carbon fractions (OC and EC), water-soluble inorganic ions and heavy metals. Total carbon (TC=OC+EC) was the most important chemical species in PE PM2.5, which means TC was the most important chemical species among these three

kinds of major components of PM2.5.

Strictly speaking, the mineral dust in PM2.5 is not directly analyzed by the instrument. It is estimated by empirical formula based on the concentration of some chemical components (mineral elements). Therefore, mineral dust cannot be regarded as the chemical composition of PM2.5. It should be considered as the source of PM2.5. On this issue, the authors will standardize the terms used in the text to avoid ambiguity. Thank you for your suggestion!

-Paragraph 705-724: First, you say that there is no non-carcinogenic risk linked with the exposure to Mn, Pb, Ni, and Zn (line 709), then you discuss the fact that the risk is much higher in the dry season (line 718). What is the point of discussing the magnitude of this risk, especially before repeating (line 723-724) that it is negligible?

Response: Although our results show that the average non-carcinogenic risk linked to heavy metals in this study was below the international thresholds, we have noticed that it has a seasonal behavior, especially for the driver group. We now first present the risk and then highlight this latter point by giving the dry/wet season ratios.

---

## Referee Comment (RC2) · Anonymous Referee #2 · 27 Nov 2018

Megacities in Africa are pollution hotspots, for which very little data have been published. Personal exposure in such environments have also received virtually no research attention. Therefore, the authors are commended for this work.

General comments: I regard the language in this paper as poor. In many instances, this prevents proper knowledge transfer to the reader and can cause misunderstanding/interpretation of the text. Additionally, it makes the paper difficult/cumbersome to read. It is not the job of the reviewer/editor to do language and/or text editing. In my opinion, this paper should not have been published in ACPD before the language and

text was acceptable. Therefore, I recommend that final review of this paper should only be considered once the language/text is improved. In the current form, too many uncertainties exist in the paper, because of the poor language.

I am also not 100% convinced that the content of this paper fits into the scope of ACP. According to the journal, ACP "... is an international scientific journal dedicated to the publication and public discussion of high-quality studies investigating the Earth's atmosphere and the underlying chemical and physical processes." My uncertainty arise from the fact that this paper is focussed more on personal exposure and not on "... underlying chemical and physical processes." Would the paper not fit better into a journal specifically considering atmospheric exposure and/or health impacts? I leave the decision to the editor. This comment should not be considered as negative in any way and it is also not a reflection of the science presented.

Specific comments: The authors must please not use the name "South West Africa" as they did in line 105, but rather keep to the term "southern West Africa", as it the rest of the paper. "South West Africa" was the name for modern-day Namibia from 1915 to 1990. I would even go so far as to recommend that the term "southern West Africa" that is abbreviated at "sWA" ("southern" in lower case) be consistently used, instead of "Southern West Africa" that is abbreviated at "SWA" ("Southern" in upper case), to ensure that the reader does not confuse the area investigated with "South West Africa" that was abbreviated at "SWA".

The author refer to "... garbage spontaneous combustion..." a couple of times. Is the garbage really combusting spontaneously, of are garbage dumps being set alight to reduce the volume of waste, to reduce pests (rats and mice) and prevent disease?

The quality of the Google Earth images presented in Figure 1 and the photos presented in Figure 2 are not good and might deteriorate further in page setting during publication (e.g. if the images are printed even smaller). I encourage the authors to ensure the best possible quality for these images.

In its current form, the paper is long. If the authors and editor agree, I would suggest that Appendixes A, B, C and D, which present the questionnaires, rather be included as supplementary material, instead of appendixes. Appendixes are published as part of the paper, while supplementary material are published separately. Readers who want to assess the content of the questionnaires can download the supplementary material, instead of the paper becoming excessively long.

I agree with Referee #1 that the authors cannot interpolate their results obtained from individuals with specific occupations and at specific locations to the wider southern West African region. All such statements should be revised.

In general, there is little comparison of the results presented in this paper to results obtained elsewhere. I appreciate that very little, if any, personal exposure data have been presented for African cities. However, even if the results presented are compared to ambient/indoor air quality results obtained in the rest of Africa (or Asia, or some other developing settings, if African reference cannot be found), the reader will be able to easier contextualise the exposure concentrations reported here. For instance, indoor air quality in semi- and informal settlements (low-income households) in South Africa (Kapwata et al., Atmosphere 2018, 9(4), 124; https://doi.org/10.3390/atmos9040124) could be compared to "Night" personal exposure of individual in this study. Also, characterisation of the plume of fire grilling of meat in an African context (Venter et al., S. Afr. J. Chem., 2015, 68, 181–194; DOI: http://dx.doi.org/10.17159/0379-4350/2015/v68a25) could possibly be compared to the exposure of woman by Domestic Fires (DF) ("grilling meat or roasting peanuts") in this study. Such comparisons will help the reader to contextualise the results presented – currently only comparing the different groups with one another does not enable the reader to contextualise the results. There might be many more references, such as the afore-mentioned, these two are just examples that I found with a quick online search.

Line 415. The author state that "The previous studies (Cao et al., 2008; Li et al., 2009; Tian et al., 2017) suggested that average OC/EC characterizes 1.1 as motor

vehicle exhaust, 2.7 as coal combustion and 9.0 as biomass burning. The OC/EC in the present study points out that biomass burning emission was the main contributor to carbonaceous aerosols for women at DF, and the mixed emissions from biomass and coal burning, even or/and motor vehicle exhaust dominated the carbonaceous aerosol sources for students at WB and drivers at MT." However, the authors should clarify these statements, since OC/EC ratio will change in a plume with aging, with the formation of secondary OC and deposition of EC. Therefore, if the above OC/EC ratios are used to characterise fresh emissions/plumes, it should be stated as such and not left to the reader to interpret.

Line 247. Fe and the heave metals reported (V, Cr, Mn, Co, Ni, Cu, Zn, Sb,473 Ba and Pb) were analysed with ED-XRF. How sure are the authors that some of the heavy metals were not part of the GM and are therefore partially double accounted for in the mass balance (Figure 5, line 493 onwards), i.e. accounted as heavy metal mass and also contributing to the mass of the GM?

Line 511. Although the authors give a citation (i.e. "Taylor and McLennan, 1985") to support the use of Fe as a tracer for geological material (GM), how accurate is this method? The authors state "Fe constitutes about 4.0% of the Earth's crust in dust of the earth's crust (Cao et al., 2005)". Are there any indications of Fe contents of local soils (and the variation on Fe contents) and how it differs from the global average value of 4%, which was used. Basically, I am asking how accurate the method is. Can the authors give any indication of accuracy? This is important, since "... it is found that GM contributed 35.8%±2.1%, 46.0%±3.7% and 42.4%±4.7% of PE PM2.5 mass concentrations for women at DF, students at WB and drivers at MT, respectively."

Line 529 "From Figure 5, evident diurnal distinguishes are observed in two major chemical compositions (OM and GM) in this study. We can see that GM exhibits the lower proportion at night (35.3%) than daytime (47.5%), indicating its close relationship with human activities." However, does meteorology not also play a role? In Line 483 the authors imply that precipitation is higher during night-time, i.e. "... spontaneous com-

bustion of waste occurs frequently during the day, because of less precipitation and higher air temperature at daytime..."

Line 539 "... due to the influence from the damp wood burning at the working time." I could not find any place where the wood moisture content was reported. Therefore, this statement and previous, as well as subsequent deductions, based on this statement, are not fact based. However, I do agree with later statements (line 566) that the wood will be damper in the wet season, i.e. "... increase in humidity (moisture content) of the wood used for grilling meat in wet season...".

Line 584. "Students at WB: nighttime PE PAHs were higher in dry season and lower in wet season compared with daytime levels, with the average D/N ratios of 0.7 and 1.8, respectively. The higher concentrations of combustion markers-BbF and BeP were observed during the day, while the higher concentrations of gasoline vehicle emission markers-DahA and BghiP were found at night (Baek et al., 1991; Wang et al., 2006), which was related to the garbage truck for waste transportation from city to the landfill during night." I am not sure that the latter explanation can be so simple, i.e. only due to "garbage truck".

Line 706. Can the authors please explain the selection of species included in the "Non-cancer risks", wherein only "four heavy metals (Mn, Ni, Zn and Pb)" were considered?

---

## Author Comment (AC2) · 17 Dec 2018

We would like to thank the reviewer for all the suggestions and comments. Below we have responded to each comment in point to point format.

[Figure]

Megacities in Africa are pollution hotspots, for which very little data have been published. Personal exposure in such environments have also received virtually no research attention. Therefore, the authors are commended for this work. General comments: I regard the language in this paper as poor. In many instances, this prevents proper knowledge transfer to the reader and can cause misunderstanding/interpretation of the text. Additionally, it makes the paper difficult/cumbersome to read. It is not the job of the reviewer/editor to do language and/or text editing. In my opinion, this paper should not have been published in ACPD before the language and text was acceptable. Therefore, I recommend that final review of this paper should only be considered once the language/text is improved. In the current form, too many uncertainties exist in the paper, because of the poor language. Response: We truly altered this issue. The revised manuscript will be proofread by a native speaker before re-submission.

I am also not 100Response: We understood the reviewer's concern and respected the decision from the editor. We would like to explain that our work is absolutely related to the PM2.5 chemical composition, emission sources and variability. This topic should be within the scope of ACP. Moreover, there are some related works on air quality conducted in sWA (same work package on Air pollution and Health) published in the ACP/AMT DACCIWA special issue, forming a coherent whole.

Specific comments: The authors must please not use the name "South West Africa" as they did in line 105, but rather keep to the term "southern West Africa", as it the rest of the paper. "South West Africa" was the name for modern-day Namibia from 1915 to 1990. I would even go so far as to recommend that the term "southern West Africa" that is abbreviated at "sWA" ("southern" in lower case) be consistently used, instead of "Southern West Africa" that is abbreviated at "SWA" ("Southern" in upper case), to ensure that the reader does not confuse the area investigated with "South West Africa" that was abbreviated at "SWA". Response: We totally agreed. The abbreviation for southern West Africa (sWA) has been revised throughout the document.

The authors refer to ". . . garbage spontaneous combustion. . ." a couple of times. Is the garbage really combusting spontaneously, or are garbage dumps being set alight to reduce the volume of waste, to reduce pests (rats and mice) and prevent disease? Response: Akouédo dump in Abidjan is a vast and old landfill. We used the term of "spontaneous combustion" as we observed several smoke plumes in the middle of the dump, far from the working area. Spontaneous combustion is a well-known phenomenon in such outdated landfill, but there was difficulty in counting its frequency of occurrence compared to control ignition. Landfill workers often burn waste when they collect trashes and recycle some of useful items. The "controlled burnt" occurs in the active part of the dump. Both processes have a high occurrence during daytime and in the dry and hot season. Here we referred waste burning in a more general manner rather than specifying spontaneous combustion.

The quality of the Google Earth images presented in Figure 1 and the photos presented in Figure 2 are not good and might deteriorate further in page setting during publication (e.g. if the images are printed even smaller). I encourage the authors to ensure the best possible quality for these images. Response: We would definitely pay more attention to the pixel issues on the figures in the revised manuscript.

In its current form, the paper is long. If the authors and editor agree, I would suggest that Appendixes A, B, C and D, which present the questionnaires, rather be included as supplementary material, instead of appendixes. Appendixes are published as part of the paper, while supplementary material are published separately. Readers who want to assess the content of the questionnaires can download the supplementary material, instead of the paper becoming excessively long. Response: We agreed with the reviewer's comment. We will move the current Appendix A-D to the supplementary material as supporting information.

I agree with Referee 1 that the authors cannot interpolate their results obtained from individuals with specific occupations and at specific locations to the wider southern West African region. All such statements should be revised. Response: Thank you for

this point. We have made the necessary changes within the text.

In general, there is little comparison of the results presented in this paper to results obtained elsewhere. I appreciate that very little, if any, personal exposure data have been presented for African cities. However, even if the results presented are compared to ambient/indoor air quality results obtained in the rest of Africa (or Asia, or some other developing settings, if African reference cannot be found), the reader will be able to easier contextualize the exposure concentrations reported here. For instance, indoor air quality in semi- and informal settlements (low-income households) in South Africa (Kapwata et al., Atmosphere 2018, 9(4), 124; https://doi.org/10.3390/atmos9040124) could be compared to "Night" personal exposure of individual in this study. Also, characterization of the plume of fire grilling of meat in an African context (Venter et al., S. Afr. J. Chem., 2015, 68, 181–194; DOI: http://dx.doi.org/10.17159/0379-4350/2015/v68a25) could possibly be compared to the exposure of woman by Domestic Fires (DF) ("grilling meat or roasting peanuts") in this study. Such comparisons will help the reader to contextualize the results presented – currently only comparing the different groups with one another does not enable the reader to contextualize the results. There might be many more references, such as the afore-mentioned, these two are just examples that I found with a quick online search. Response: We understood this point of view. The citations on our work are related to either ambient concentrations or source characterization. The suggested references should not be applicable. Venter et al. (2015) presented a study on charcoal combustion with a specific type of barbecue, totally different from the conditions at DF (wood in barrel). Kapwata et al. (2018) showed the work on PM4 (dissimilar with our PM2.5) and no chemical composition was provided. In lines 370 and 384 of our original manuscript, we firstly compared the average PE PM2.5 levels to the weekly ambient PM2.5 concentrations obtained in the same area and similar sampling period, and also compared the daytime PE PM2.5 mass concentrations with the daytime ambient PM2.5 in the same area and exactly the same sampling dates (Djossou et al., 2018). We also used the results of the PAHs exposure measured in Cotonou in a previous study (Fanou et al., 2006). We provide

additional references of previous works on personal exposure to PM2.5 in household in Tanzania (Titcombe and Simcik, 2015) and for students in Ghana (Arku et al., 2014). And we compared the results with our data in this study as follows: "The 5-h PM2.5 average personal exposure concentration was 1574 $\mu$g m-3 ($\pm$287, n = 3) for open wood fires in households in the Njombe district of Tanzania (Titcombe and Simcik, 2015), and was comparable to the highest 12-h exposure level to PM2.5 for women at DF site in this study (1164.7 $\mu$g m-3, daytime in wet season, July 5th), and was 4.7 times of the daily average PE PM2.5 concentration in dry and wet seasons (331.7$\pm$190.7 $\mu$g m-3)." "In the study of Titcombe and Simcik (2015), we also found that the 5-h average total PAH personal exposure concentration was 5040 ng m-3 ($\pm$909, n = 3) for open wood fires in households in the Njombe district of Tanzania, which was much higher ( 65 times) than the women exposure PAHs at DF site in the current research. The highest 12-h exposure PAHs for women at DF site in this study was 469.7 ng m-3 (daytime in wet season, July 6th), approximately one-tenth of the PAHs concentration from open wood fires in Tanzania mentioned above. The large PE PAH concentrations difference between these two studies may be influenced by many factors such as wood type, combustion state, stove structure and sampling time." "Student (10-17 years old) PM2.5 exposures ranged from less than 10 $\mu$g m-3 to more than 150 $\mu$g m-3 (mean 56 $\mu$g m-3) in four neighborhoods in Accra, Ghana (Arku et al., 2014), much lower than that for students at WB site (356.9$\pm$71.9 $\mu$g m-3). It can be seen that the high exposure of students in this study is likely to be related to the waste burning emissions, while there was no obvious strong PM2.5 emission source in the study of Arku et al. (2014)." We have also added the comparisons as mentioned in the revised manuscript. Reference: Arku, R. E., Dionisio, K. L., Hughes, A. F., Vallarino, J., Spengler, J. D., Castro, M. C., Agyei-Mensah, S., and Ezzati, M.: Personal particulate matter exposures and locations of students in four neighborhoods in Accra, Ghana. J. Expo. Sci. Environ. Epidemiol., 1–10, 2014. Titcombe, M. E., and Simcik, M.: Personal and indoor exposure to PM2.5 and polycyclic aromatic hydrocarbons in the southern highlands of Tanzania: a pilot-scale study. Environ. Monit. Assess., 180, 461–476, 2011.

[Figure]

Line 415. The author state that "The previous studies (Cao et al., 2008; Li et al., 2009; Tian et al., 2017) suggested that average OC/EC characterizes 1.1 as motor vehicle exhaust, 2.7 as coal combustion and 9.0 as biomass burning. The OC/EC in the present study points out that biomass burning emission was the main contributor to carbonaceous aerosols for women at DF, and the mixed emissions from biomass and coal burning, even or/and motor vehicle exhaust dominated the carbonaceous aerosol sources for students at WB and drivers at MT." However, the authors should clarify these statements, since OC/EC ratio will change in a plume with aging, with the formation of secondary OC and deposition of EC. Therefore, if the above OC/EC ratios are used to characterize fresh emissions/plumes, it should be stated as such and not left to the reader to interpret. Response: The statements have been revised as follows: "The previous studies (Cachier et al., 1989; Cao et al., 2005a; Cao et al., 2008; Li et al., 2009; Tian et al., 2017; Watson et al., 2001) suggested that average OC/EC characterizes 1.1 as motor vehicle exhaust, 2.7 as coal combustion and 9.0 as biomass burning for source samples (fresh emissions/plumes)". The reason why we could compare the OC/EC results of the personal exposure data in this study with the above source samples is because that the participants were close or around to the typical anthropogenic sources in this study. In addition, PM2.5 emitted from the pollution sources still maintained a relatively fresh state (less aging) without long-distance transport, and then was inhaled into human body. Therefore, the OC/EC ratio comparison results in this study could yield reliable conclusions as described in the texts (originally lines 418-421). Reference: Cachier, H., Bremond, M. P., and Buat-Menard, P.: Carbonaceous aerosols from different tropical biomass burning sources. Nature, 340, 371–373, 1989. Cao, J. J., Wu, F., Chow, J. C., Lee, S. C., Li, Y., Chen, S. W., An, Z. S., Fung, K. K., Watson, J. G., Zhu, C. S., and Liu, S. X.: Characterization and source apportionment of atmospheric organic and elemental carbon during fall and winter of 2003 in Xi'an, China. Atmos. Chem. Phys., 5, 3127–3137, 2005a. Watson, J. G., Chow, J. C., and Houck, J. E.: PM2.5 chemical source profiles for vehicle exhaust, vegetative burning, geological material, and coal burning in northwestern Colorado during 1995.

Chemosphere, 43, 1141–1151, 2001.

Line 247. Fe and the heave metals reported (V, Cr, Mn, Co, Ni, Cu, Zn, Sb, Ba and Pb) were analyzed with ED-XRF. How sure are the authors that some of the heavy metals were not part of the GM and are therefore partially double accounted for in the mass balance (Figure 5, line 493 onwards), i.e. accounted as heavy metal mass and also contributing to the mass of the GM? Response: The abundance of Fe in the earth's crust is about 4

Line 511. Although the authors give a citation (i.e. "Taylor and McLennan, 1985") to support the use of Fe as a tracer for geological material (GM), how accurate is this method? The authors state "Fe constitutes about 4.0Response: Fe is the most important metal and one of the major constituents of the lithosphere. Its average content of the Earth's crust is about 5Because of the lack of elemental composition obtained in topsoil of Africa, the authors used 4.0Reference: Hao, Y. C., Guo, Z. G., Yang, Z. S., Fang, M., and Feng, J. L.: Seasonal variations and sources of various elements in the atmospheric aerosols in Qingdao, China, Atmos. Res., 85, 27-37, 2007. Kabata-Pendias, A., and Mukherjee, A. B.: Trace Elements from Soil to Human. Springer-Verlag, Berlin Heidelberg, Germany, pp. 381-393, 2007. Sun, Y. Y., Hu, X., Wu, J. C., Lian, H. Z., and Chen, Y. J.: Fractionation and health risks of atmospheric particle-bound As and heavy metals in summer and winter, Sci. Total Environ., 493, 487-494, 2014. Wu, F., Zhang, D. Z., Cao, J. J., Xu, H. M., and An, Z. S.: Soil-derived sulfate in atmospheric dust particles at Taklimakan desert, Geophy. Res. Lett., 39, L24803, doi:10.1029/2012GL054406, 2012.

Line 529 "From Figure 5, evident diurnal distinguishes are observed in two major chemical compositions (OM and GM) in this study. We can see that GM exhibits the lower proportion at night (35.3Response: As our best knowledge, the main source of the geological material in PM2.5 is from crust dust. In addition to the effects of crust erosion by water and wind, it is largely related to the physical activities of human, such as resuspension from individual activities, construction activities and etc. The second reason (meteorological factor) shown in original line 483 explained the day-night ratio of heavy metals in PM2.5, that is, the absolute concentrations of heavy metals; while the content in original line 529 discussed the proportion of geological material in PM2.5. The meteorological factors, including precipitation, indeed cause scouring action on particulate matters (i.e., wet deposition), but they have little effect on altering the proportion of geological material in PM2.5. Therefore, lower geological material proportion in PM2.5 at night in this study was mainly due to the resuspension of the geological material from the less individual activities.

Line 539 ". . . due to the influence from the damp wood burning at the working time." I could not find any place where the wood moisture content was reported. Therefore, this statement and previous, as well as subsequent deductions, based on this statement, are not fact based. However, I do agree with later statements (line 566) that the wood will be damper in the wet season, i.e. ". . . increase in humidity (moisture content) of the wood used for grilling meat in wet season. . .". Response: In this study, we did not measure the moisture of the wood used for barbecue in women's work. Since the grilling fuel (wood) was placed in the open area (no shield or roof), the abundant rainfall in wet season inevitably led to moisture increase in the wood. With on-field observation, it frequently took a long time to ignite the wood, which emitted more plume from damp fuels in the wet season during the sampling period. In previous studies (Chen et al., 2010; Grandesso et al., 2011; Keita et al., 2018; Shen et al., 2012, 2013), the results showed that biomass fuel with high moisture often required additional energy to vaporize the water and hence resulted in low combustion efficiency and high pollutant emissions. The emission factor (EF) of OC increase with the fuel moisture content (Chen et al., 2010). The authors have modified the original statement in lines 538-541 as follows: "There is an exception in the last case, i.e., OM proportion at daytime women PE PM2.5 was much higher (50.8Reference: Chen, L.-W. A., Verburg, P., Shackelford, A., Zhu, D., Susfalk, R., Chow, J. C., and Watson J. G.: Moisture effects on carbon and nitrogen emission from burning of wildland biomass. Atmos. Chem. Phys., 10, 6617–6625, 2010. Grandesso, E., Gullett, B., Touati, A., and Tabor,

D.: Effect of moisture, charge size, and chlorine concentration on PCDD/F emissions from simulated open burning of forest biomass. Environ. Sci. Technol., 45, 3887–3894, 2011. Shen, G., Wei, S., Wei, W., Zhang, Y., Min, Y., Wang, B., Wang, R., Li, W., Shen, H., Huang, Y., Yang, Y., Wang, W., Wang, X., Wang, X., and Tao, S.: Emission factors, size distributions, and emission inventories of carbonaceous particulate matter from residential wood combustion in rural China. Environ. Sci. Technol., 46, 4207−4214, 2012.

Line 584. "Students at WB: nighttime PE PAHs were higher in dry season and lower in wet season compared with daytime levels, with the average D/N ratios of 0.7 and 1.8, respectively. The higher concentrations of combustion markers-BbF and BeP were observed during the day, while the higher concentrations of gasoline vehicle emission markers-DahA and BghiP were found at night (Baek et al., 1991; Wang et al., 2006), which was related to the garbage truck for waste transportation from city to the land-fill during night." I am not sure that the latter explanation can be so simple, i.e. only due to "garbage truck". Response: We apologize for the misleading. Referring to the middle panel of Figure 6A, it shows that the distributions of PAHs in students' PM2.5 personal exposure samples in the dry and wet seasons were basically the same, with the higher concentrations in the dry season. The authors discussed the diurnal variation of PAHs concentration in different seasons and the dominant PAH species in daytime and nighttime. Both PAH profiles had a similar feature of high combustion markers of BbF and BeP, and gasoline emission markers of DahA and BghiP. Besides, with the large error bars (standard deviations) of PAH concentrations shown in this Figure, we believe that the previous statements about the dominant PAHs species in the daytime and nighttime are not so supportive, and thus the statements have been deleted from the revised manuscript. In addition, the statements in lines 586-591 have been revised as following: "Both of the PAH profiles in the day and night were with the features of higher combustion markers-BbF and BeP, and higher gasoline vehicle emission markers-DahA and BghiP (Baek et al., 1991; Wang et al., 2006)".

Line 706. Can the authors please explain the selection of species included in the "Non-cancer risks", wherein only "four heavy metals (Mn, Ni, Zn and Pb)" were considered? Response: The non-carcinogenic risks of heavy metals in PM2.5 via inhalation were calculated according to the U.S. EPA health risk assessment model (USEPA, 2004, 2011). U.S. EPA health risk assessment model is the process to estimate the nature and probability of adverse health effects in humans who may be exposed to chemicals in contaminated environmental media, now or in the future. The reason we only chose Mn, Ni, Zn and Pb to assess the health risks in personal exposure PM2.5 samples is because these four metals (among all the chemicals we analyzed in this study) are included in this U.S. EPA health risk assessment model and they are assessed to be hazardous to human health in the previous studies (e.g., Hu et al., 2018; Kong et al., 2015; Sun et al., 2014; Xu et al., 2018).

Please also note the supplement to this comment:
https://www.atmos-chem-phys-discuss.net/acp-2018-1060/acp-2018-1060-AC2-supplement.pdf
* * *

---

## Author Response (AR1)

We would like to thank the reviewer for both comments and suggestions on our manuscript. We have addressed and responded to each comment below.

Comments and suggestions for improvement:

- I understand that the authors are not native speakers but in some parts of the manuscript, the clumsy phrasing hinders comprehension. This point should be taken care of.

Response: The manuscript has been proofread by a native speaker.

- In the chemical analysis, I am surprised by the choice of Fe as a tracer of the crustal component of the aerosol. It is well known that at least a part of its concentration is contributed by anthropogenic activities. Wouldn't Al or Ca be a better choice? By the way, why were these elements not quantified by the XRF analysis?

Response: This is a good point. In order to analyze the carbonaceous aerosol, the authors selected quartz fiber filters to collect personal exposure PM$_{2.5}$ samples in this study. Due to the limitations of personal exposure sampling, it is difficult to collect both quartz and Teflon filter samples simultaneously.

Moreover, the analytical uncertainties by using ED-XRF for small molecular weight crustal elements in quartz fiber filter (due to high background for Na, Al, Ca and Mg), such as Al, Si and Ca, are high. So, Al, Si and Ca are not suitable to be used as a tracer for the crustal component of the aerosol in this study. Meanwhile, the high accuracy of Fe analysis with ED-XRF has been demonstrated in our previous publication (Xu et al., 2016b), and it has been often used as a tracer for crustal component in PM$_{2.5}$ (e.g., Cao et al., 2005b; Hao et al., 2007; Sun et al., 2014; Wu et al., 2012; Xu et al., 2016b).

Furthermore, based on the previous references (Gelado-Caballero et al., 2012; Zhuang et al., 2001), the enrichment factors of Fe in dust storm period and non-dust storm were both 1-2, always < 10, proving that Fe in aerosol was still mainly derived from the crustal source. Therefore, taking into account the above points, the authors finally picked Fe as a tracer of the crustal component in this study.

*Reference:*

Cao, J. J., Rong, B., Lee, S. C., Chow, J. C., Ho, K. F., Liu, S. X., and Zhu, C. S.: Composition of indoor aerosols at emperor Qin's terra-cotta museum, Xi'an, China, during summer, China Part., 3(3), 170-175, 2005b.

Gelado-Caballero, M. D., López-García, P., Prieto, S., Patey, M. D., Collado, C., and Hérnández-Brito, J. J.: Long-term aerosol measurements in Gran Canaria, Canary Islands: Particle concentration, sources and elemental composition, J. Geophy. Res.-Atmos., 117, D03304, doi:10.1029/2011JD016646, 2012.

Hao, Y. C., Guo, Z. G., Yang, Z. S., Fang, M., and Feng, J. L.: Seasonal variations and sources of various elements in the atmospheric aerosols in Qingdao, China, Atmos. Res., 85, 27-37, 2007.

Sun, Y. Y., Hu, X., Wu, J. C., Lian, H. Z., and Chen, Y. J.: Fractionation and health risks of atmospheric particle-bound As and heavy metals in summer and winter, Sci. Total Environ., 493, 487-494, 2014.

Wu, F., Zhang, D. Z., Cao, J. J., Xu, H. M., and An, Z.S: Soil-derived sulfate in atmospheric dust particles at Taklimakan desert, Geophy. Res. Lett., 39, L24803, doi:10.1029/2012GL054406, 2012.

Xu, H.M., Cao, J. J., Chow, J. C., Huang, R.-J., Shen, Z. X., Chen, L. W. A., Ho, K. F., and Watson, J. G..: Inter-annual variability of wintertime $PM_{2.5}$ chemical composition in Xi'an, China: Evidences of changing source emissions, Sci. Total Environ. 545-546, 546-555, 2016b.

Zhuang, G. S., Guo, J. H., Yuan, H., and Zhao, C. Y.: The compositions, sources, and size distribution of the dust storm from China in spring of 2000 and its impact on the global environment, Chinese Sci. Bull., 46(11), 895-900, 2001.

- In the health risk assessment, it would be useful to detail the type of risk quantified. The categories 'cancer risk' and 'non-cancer risk' are very broad. Also, is the risk a long-term or a short-term one? Why did you assess only the risks resulting from exposure to Mn, Ni, Zn, Pb, the PAHs and the PAEs? There is also a risk due to exposure to $PM_{2.5}$ and given the large concentrations reported in your work, I expect this one might be very important.

Response: We do agree and understand the reviewer's concern. The details of these categories have been clarified and revised in the manuscript:

"*The heavy metals non-carcinogenic risks and toxic organics carcinogenic risks of $PM_{2.5}$ via inhalation were calculated according to the U.S. EPA health risk assessment model (USEPA, 2004, 2011).*"

U.S. EPA health risk assessment model is the process to estimate the nature and probability of adverse health effects in humans who may be exposed to chemicals in contaminated environmental media, now or in the future. The reason for choosing Mn, Ni, Zn, Pb, PAHs and PAEs to assess the health risks in personal exposure $PM_{2.5}$ samples is because these chemicals (among all the chemicals we analyzed in this study) are included in this model and they are assessed to be hazardous to human health in the previous studies (e.g., Hu et al., 2018; Kong et al., 2015; Sun et al., 2014; Xu et al.,

2018a).

Moreover, indeed, as the reviewer said "There is also a risk due to exposure to PM$_{2.5}$", but PM$_{2.5}$ is a complex mixture containing a lot of chemicals. There is no clear and better way to assess its whole health risks for now based on PM$_{2.5}$ chemical concentrations (except for the model simulation and medical animal exposure experiments). So, at this moment, we calculated the risks of the certain toxic chemicals in PM$_{2.5}$ to estimate PM$_{2.5}$ health risks.

> *Reference:*
> *Hu, Y. J., Bao, L. J., Huang, C. L., Li, S. M., Li, Liu, P., and Zeng, E. Y.: Assessment of airborne polycyclic aromatic hydrocarbons in a megacity of South China: Spatiotemporal variability, indoor-outdoor interplay and potential human health risk, Environ. Pollut., 238, 431-439, 2018.*
> *Kong, S. F., Li, L., Li, X. X., Yin, Y., Chen, K., Liu, D. T., Yuan, L., Zhang, Y. J., Shan, Y. P., and Ji, Y. Q.: The impacts of firework burning at the Chinese Spring Festival on air quality: insights of tracers, source evolution and aging processes, Atmos. Chem. Phys., 15, 2167-2184, 2015.*
> *Sun, Y. Y., Hu, X., Wu, J. C., Lian, H. Z., and Chen, Y. J.: Fractionation and health risks of atmospheric particle-bound As and heavy metals in summer and winter, Sci. Total Environ., 493, 487-494, 2014.*
> *Xu, H. M., Guinot, B., Cao, J. J., Li, Y. Q., Niu, X. Y., Ho, K. F., Shen, Z. X., Liu, S. X., Zhang, T., Lei, Y. L., Zhang, Q., Sun, J., and Gao, J. J.: Source, health risk and composition impact of outdoor very fine particles (VFPs) to school indoor environment in Xi'an, Northwestern China, Sci. Total Environ., 612, 238-246, 2018a.*

- In the results section (line 328-330), you cannot extrapolate to the whole SWA region your results collected during two weeks at three very specific locations.

Response: Thank you for pointing out. The authors have revised this statement to:

> "*The average PE PM$_{2.5}$ mass concentrations were 331.7±190.7, 356.9±71.9 and 242.8±67.6 µg m$^{-3}$ for women at Domestic Fires (DF), students at Waste Burning (WB) and drivers at Motorcycle Traffic (MT), respectively, in this study.*"

Moreover, the authors have checked related issue and made corresponding changes throughout the revised manuscript.

- Line 400: you say that total carbon was the most important chemical species in PE PM$_{2.5}$ but it contributes only about 20% to the mass concentration. Isn't this contradictory? What about mineral dust?

Response: We are sorry for the confusion. In section "3.1.2. PE PM$_{2.5}$ chemical compositions", the authors were talking about the PE PM$_{2.5}$ chemical compositions, which include carbon fractions (OC and EC), water-soluble inorganic ions and heavy metals. Total carbon (TC=OC+EC) was the most important chemical species in PE $PM_{2.5}$, which means TC was the most important chemical species among these three kinds of major components of $PM_{2.5}$.

Strictly speaking, the mineral dust in $PM_{2.5}$ is not directly analyzed by the instrument. It is estimated by empirical formula based on the concentration of some chemical components (mineral elements). Therefore, mineral dust cannot be regarded as the chemical composition of $PM_{2.5}$. It should be considered as the source of $PM_{2.5}$. On this issue, the authors will standardize the terms used in the text to avoid ambiguity. Thank you for your suggestion!

-Paragraph 705-724: First, you say that there is no non-carcinogenic risk linked with the exposure to Mn, Pb, Ni, and Zn (line 709), then you discuss the fact that the risk is much higher in the dry season (line 718). What is the point of discussing the magnitude of this risk, especially before repeating (line 723-724) that it is negligible?

Response: Although our results show that the average non-carcinogenic risk linked to heavy metals in this study was below the international thresholds, we have noticed that it has a seasonal behavior, especially for the driver group. We now first present the risk and then highlight this latter point by giving the dry/wet season ratios.

**Anonymous Referee #2**

We would like to thank the reviewer for all the suggestions and comments. Below we have responded to each comment in point to point format.

Megacities in Africa are pollution hotspots, for which very little data have been published. Personal exposure in such environments have also received virtually no research attention. Therefore, the authors are commended for this work.

General comments: I regard the language in this paper as poor. In many instances, this prevents proper knowledge transfer to the reader and can cause misunderstanding/interpretation of the text. Additionally, it makes the paper difficult/cumbersome to read. It is not the job of the reviewer/editor to do language and/or text editing. In my opinion, this paper should not have been published in ACPD before the language and text was acceptable. Therefore, I recommend that final review of this paper should only be considered once the language/text is improved. In the current form, too many uncertainties exist in the paper, because of the poor language.

Response: The manuscript has been revised by a native speaker before re-submission.

I am also not 100% convinced that the content of this paper fits into the scope of ACP. According to the journal, ACP ". . . is an international scientific journal dedicated to the publication and public discussion of high-quality studies investigating the Earth's atmosphere and the underlying chemical and physical processes." My uncertainty arises from the fact that this paper focused more on personal exposure and not on ". . . underlying chemical and physical processes." Would the paper not fit better into a journal specifically considering atmospheric exposure and/or health impacts? I leave the decision to the editor. This comment should not be considered as negative in any way and it is also not a reflection of the science presented.

Response: We understood the reviewer's concern and respected the decision from the editor. We would like to explain that our work is absolutely related to the $PM_{2.5}$ chemical composition, emission sources and variability. This topic should be within the scope of ACP. Moreover, there are some related works on air quality conducted in sWA (same work package on Air pollution and Health) published in the ACP/AMT DACCIWA special issue, forming a coherent whole.

Specific comments: The authors must please not use the name "South West Africa" as they did in line 105, but rather keep to the term "southern West Africa", as it the rest of the paper. "South West Africa" was the name for modern-day Namibia from 1915 to

1990. I would even go so far as to recommend that the term "southern West Africa" that is abbreviated at "sWA" ("southern" in lower case) be consistently used, instead of "Southern West Africa" that is abbreviated at "SWA" ("Southern" in upper case), to ensure that the reader does not confuse the area investigated with "South West Africa" that was abbreviated at "SWA".

Response: We totally agreed. The abbreviation for southern West Africa (sWA) has been revised throughout the document.

The authors refer to ". . . garbage spontaneous combustion. . ." a couple of times. Is the garbage really combusting spontaneously, or are garbage dumps being set alight to reduce the volume of waste, to reduce pests (rats and mice) and prevent disease?

Response: Akouédo dump in Abidjan is a vast and old landfill. We used the term of "spontaneous combustion" as we observed several smoke plumes in the middle of the dump, far from the working area. Spontaneous combustion is a well-known phenomenon in such outdated landfill, but there was difficulty in counting its frequency of occurrence compared to control ignition. Landfill workers often burn waste when they collect trashes and recycle some of useful items. The "controlled burnt" occurs in the active part of the dump. Both processes have a high occurrence during daytime and in the dry and hot season. Here we referred waste burning in a more general manner rather than specifying spontaneous combustion.

The quality of the Google Earth images presented in Figure 1 and the photos presented in Figure 2 are not good and might deteriorate further in page setting during publication (e.g. if the images are printed even smaller). I encourage the authors to ensure the best possible quality for these images.

Response: We definitely payed more attention to the pixel issues on the figures in the revised manuscript.

In its current form, the paper is long. If the authors and editor agree, I would suggest that Appendixes A, B, C and D, which present the questionnaires, rather be included as supplementary material, instead of appendixes. Appendixes are published as part of the paper, while supplementary material are published separately. Readers who want to assess the content of the questionnaires can download the supplementary material, instead of the paper becoming excessively long.

Response: We agreed with the reviewer's comment. We have moved the current Appendix A-D to the supplementary material as supporting information (SI A-D).

I agree with Referee #1 that the authors cannot interpolate their results obtained from individuals with specific occupations and at specific locations to the wider southern West African region. All such statements should be revised.

Response: Thank you for this point. We have made the necessary changes within the text.

In general, there is little comparison of the results presented in this paper to results obtained elsewhere. I appreciate that very little, if any, personal exposure data have been presented for African cities. However, even if the results presented are compared to ambient/indoor air quality results obtained in the rest of Africa (or Asia, or some other developing settings, if African reference cannot be found), the reader will be able to easier contextualize the exposure concentrations reported here. For instance, indoor air quality in semi- and informal settlements (low-income households) in South Africa (Kapwata et al., Atmosphere 2018, 9(4), 124; https://doi.org/10.3390/atmos9040124) could be compared to "Night" personal exposure of individual in this study. Also, characterization of the plume of fire grilling of meat in an African context (Venter et al., S. Afr. J. Chem., 2015, 68, 181–194; DOI: http://dx.doi.org/10.17159/0379-4350/2015/v68a25) could possibly be compared to the exposure of woman by Domestic Fires (DF) ("grilling meat or roasting peanuts") in this study. Such comparisons will help the reader to contextualize the results presented – currently only comparing the different groups with one another does not enable the reader to contextualize the results. There might be many more references, such as the afore-mentioned, these two are just examples that I found with a quick online search.

Response: We understood this point of view. The citations on our work are related to either ambient concentrations or source characterization. The suggested references should not be applicable. Venter et al. (2015) presented a study on charcoal combustion with a specific type of barbecue, totally different from the conditions at DF (wood in barrel). Kapwata et al. (2018) showed the work on $PM_4$ (dissimilar with our $PM_{2.5}$) and no chemical composition was provided.

In lines 370 and 384 of our original manuscript, we firstly compared the average PE $PM_{2.5}$ levels to the weekly ambient $PM_{2.5}$ concentrations obtained in the same area and similar sampling period, and also compared the daytime PE $PM_{2.5}$ mass concentrations with the daytime ambient $PM_{2.5}$ in the same area and exactly the same sampling dates (Djossou et al., 2018). We also used the results of the PAHs exposure measured in Cotonou in a previous study (Fanou et al., 2006). We provide additional references of previous works on personal exposure to $PM_{2.5}$ in household in Tanzania (Titcombe and Simcik, 2011) and for students in Ghana (Arku et al., 2014). And we compared the results with our data in this study as follows:

> "The 5-h $PM_{2.5}$ average personal exposure concentration was 1574 μg m$^{-3}$ (±287, n = 3) for open wood fires in households in the Njombe district of Tanzania (Titcombe and Simcik, 2011), and was comparable to the highest 12-h exposure level to $PM_{2.5}$ for women at DF site in this study (1164.7 μg m$^{-3}$, daytime in wet

*season, July 5th), and was 4.7 times of the daily average PE PM$_{2.5}$ concentration in dry and wet seasons (331.7±190.7 µg m$^{-3}$).”*

*“In the study of Titcombe and Simcik (2011), the authors found that the 5-h average total PAH personal exposure concentration was 5040 ng m$^{-3}$ (±909, n = 3) for open wood fires in households in the Njombe district of Tanzania, which was much higher (~65 times) than the women exposure PAHs at DF site in the current research. The highest 12-h exposure PAHs for women at DF site in this study was 469.7 ng m$^{-3}$ (daytime in wet season, July 6th), approximately one-tenth of the PAHs concentration from open wood fires in Tanzania mentioned above. The large PE PAH concentrations difference between these two studies may be influenced by many factors such as wood type, combustion state, stove structure and sampling time.”*

*“Student (10-17 years old) PM$_{2.5}$ exposures ranged from less than 10 µg m$^{-3}$ to more than 150 µg m$^{-3}$ (mean 56 µg m$^{-3}$) in four neighborhoods in Accra, Ghana (Arku et al., 2014), much lower than that for students at WB site (356.9±71.9 µg m$^{-3}$). It can be seen that the high exposure of students in this study is likely to be related to the waste burning emissions, while there was no obvious strong PM$_{2.5}$ emission source in the study of Arku et al. (2014).”*

We have also added the comparisons as mentioned above in the revised manuscript.

Line 511. Although the authors give a citation (i.e. "Taylor and McLennan, 1985") to support the use of Fe as a tracer for geological material (GM), how accurate is this method? The authors state "Fe constitutes about 4.0% of the Earth's crust in dust of the earth's crust (Cao et al., 2005)". Are there any indications of Fe contents of local soils (and the variation on Fe contents) and how it differs from the global average value of 4%, which was used? Basically, I am asking how accurate the method is. Can the authors give any indication of accuracy? This is important, since ". . . it is found that GM contributed 35.8%±2.1%, 46.0%±3.7% and 42.4%±4.7% of PE $PM_{2.5}$ mass concentrations for women at DF, students at WB and drivers at MT, respectively."

Response: Fe is the most important metal and one of the major constituents of the lithosphere. Its average content of the Earth's crust is about 5%. The global abundance of Fe is around 4.5% (Kabata-Pendias and Mukherjee, 2007). The typical range of Fe contents in soils is between 0.1 and 10% and its distribution in soil is variable, which is controlled by several soil parameters (Kabata-Pendias and Mukherjee, 2007). The authoritative study from Taylor and McLennan (1985) of elemental content in soils showed that the global abundance of Fe is around 3.5%.

Because of the lack of elemental composition obtained in topsoil of Africa, the authors used 4.0% (global mean Fe content from the mentioned literatures) as a percentage of Fe in the topsoil of African study area for geological material estimation in this study. The value was widely used in other literatures (Cao et al., 2005b; Hao et al., 2007; Sun et al., 2014; Wu et al., 2012; Xu et al., 2016b). In our opinion, this method can roughly indicate the contribution of the geological material to atmospheric particle matters to a certain extent. Moreover, since the geological material of three sampling sites in this study were estimated using a consistent method, the relative results were comparable.

Line 529 "From Figure 5, evident diurnal distinguishes are observed in two major chemical compositions (OM and GM) in this study. We can see that GM exhibits the lower proportion at night (35.3%) than daytime (47.5%), indicating its close relationship with human activities." However, does meteorology not also play a role? In Line 483 the authors imply that precipitation is higher during night-time, i.e. ". . . spontaneous combustion of waste occurs frequently during the day, because of less precipitation and higher air temperature at daytime. . ."

Response: As our best knowledge, the main source of the geological material in $PM_{2.5}$ is from crust dust. In addition to the effects of crust erosion by water and wind, it is largely related to the physical activities of human, such as resuspension from individual activities, construction activities and etc. The second reason (meteorological factor) shown in original line 483 explained the day-night ratio of heavy metals in $PM_{2.5}$, that is, the absolute concentrations of heavy metals; while the content in original line 529 discussed the proportion of geological material in $PM_{2.5}$. The meteorological factors, including precipitation, indeed cause scouring action on particulate matters (i.e., wet deposition), but they have little effect on altering the proportion of geological material in $PM_{2.5}$. Therefore, lower geological material proportion in $PM_{2.5}$ at night in this study was mainly due to the resuspension of the geological material from the less individual activities.

Line 539 ". . . due to the influence from the damp wood burning at the working time." I could not find any place where the wood moisture content was reported. Therefore, this statement and previous, as well as subsequent deductions, based on this statement, are not fact based. However, I do agree with later statements (line 566) that the wood will be damper in the wet season, i.e. ". . . increase in humidity (moisture content) of the wood used for grilling meat in wet season. . .".

Response: In this study, we did not measure the moisture of the wood used for barbecue in women's work. Since the grilling fuel (wood) was placed in the open area (no shield or roof), the abundant rainfall in wet season inevitably led to moisture increase in the wood. With on-field observation, it frequently took a long time to ignite the wood, which emitted more plume from damp fuels in the wet season during the sampling period. In previous studies (Chen et al., 2010; Grandesso et al., 2011; Keita et al., 2018; Shen et al., 2012, 2013), the results showed that biomass fuel with high moisture often required additional energy to vaporize the water and hence resulted in low combustion efficiency and high pollutant emissions. The emission factor (EF) of OC increase with the fuel moisture content (Chen et al., 2010).

The authors have modified the original statement in lines 538-541 as follows:

> *"An exception is that OM proportion of women PE $PM_{2.5}$ at daytime (50.8%) was much higher than nighttime (38.2%) in wet season, due to the influences from the damp wood burning at the working time. Burning biomass fuel with high moisture often results in low combustion efficiency, long smoldering period and high air pollutant emissions (Grandesso et al., 2011; Shen et al., 2012, 2013). The emission factor of OC usually increases with the fuel moisture content (Chen et al., 2010; Keita et al., 2018). Therefore, burning the damp wood led to higher OC emission than dry wood, in-line with the observation for women PE results in this study."*

> *Reference:*
> *Chen, L.-W. A., Verburg, P., Shackelford, A., Zhu, D., Susfalk, R., Chow, J. C., and Watson J. G.: Moisture effects on carbon and nitrogen emission from burning of wildland biomass. Atmos. Chem. Phys., 10, 6617–6625, 2010.*
> *Grandesso, E., Gullett, B., Touati, A., and Tabor, D.: Effect of moisture, charge size, and chlorine concentration on PCDD/F emissions from simulated open*

*burning of forest biomass. Environ. Sci. Technol., 45, 3887–3894, 2011.*

*Shen, G., Wei, S., Wei, W., Zhang, Y., Min, Y., Wang, B., Wang, R., Li, W., Shen, H., Huang, Y., Yang, Y., Wang, W., Wang, X., Wang, X., and Tao, S.: Emission factors, size distributions, and emission inventories of carbonaceous particulate matter from residential wood combustion in rural China. Environ. Sci. Technol., 46, 4207−4214, 2012.*

Line 584. "Students at WB: nighttime PE PAHs were higher in dry season and lower in wet season compared with daytime levels, with the average D/N ratios of 0.7 and 1.8, respectively. The higher concentrations of combustion markers-BbF and BeP were observed during the day, while the higher concentrations of gasoline vehicle emission markers-DahA and BghiP were found at night (Baek et al., 1991; Wang et al., 2006), which was related to the garbage truck for waste transportation from city to the landfill during night." I am not sure that the latter explanation can be so simple, i.e. only due to "garbage truck".

Response: We apologized for the misleading. Referring to the middle panel of Figure 6A, it shows that the distributions of PAHs in students' $PM_{2.5}$ personal exposure samples in the dry and wet seasons were basically the same, with the higher concentrations in the dry season. The authors discussed the diurnal variation of PAHs concentration in different seasons and the dominant PAH species in daytime and nighttime. Both PAH profiles had a similar feature of high combustion markers of BbF and BeP, and gasoline emission markers of DahA and BghiP. Besides, with the large error bars (standard deviations) of PAH concentrations shown in this Figure, we believe that the previous statements about the dominant PAHs species in the daytime and nighttime are not so supportive, and thus the statements have been deleted from the revised manuscript.

In addition, the statements in lines 586-591 have been revised as following:

*"Both the PAH profiles were featured with high combustion markers of BbF and benzo[e]pyrene (BeP), and high gasoline vehicle emission markers of dibenzo[a,h]anthracene (DahA) and BghiP (Baek et al., 1991; Wang et al., 2006)."*

Line 706. Can the authors please explain the selection of species included in the "Noncancer risks", wherein only "four heavy metals (Mn, Ni, Zn and Pb)" were considered?

Response: The non-carcinogenic risks of heavy metals in $PM_{2.5}$ via inhalation were calculated according to the U.S. EPA health risk assessment model (USEPA, 2004, 2011). U.S. EPA health risk assessment model is the process to estimate the nature and probability of adverse health effects in humans who may be exposed to chemicals in contaminated environmental media, now or in the future. The reason we only chose Mn, Ni, Zn and Pb to assess the health risks in personal exposure $PM_{2.5}$ samples is because these four metals (among all the chemicals we analyzed in this study) are included in this U.S. EPA health risk assessment model and they are assessed to be hazardous to human health in the previous studies (e.g., Hu et al., 2018; Kong et al., 2015; Sun et al., 2014; Xu et al., 2018a).

[revised manuscript text omitted]